# Hypercapnic warm-up and re-warm-up–A novel experimental approach in swimming sprint

Natalia Danek[1]*, Stefan Szczepan[2], Zofia Wróblewska[3], Kamil Michalik[4], Marek Zatoń[1]

1 Department of Physiology and Biochemistry, Faculty of Physical Education and Sport Science, Wroclaw University of Health and Sport Sciences, Wroclaw, Poland, 2 Department of Swimming, Faculty of Physical Education and Sport Science, Wroclaw University of Health and Sport Sciences, Wroclaw, Poland, 3 Faculty of Pure and Applied Mathematics, Wroclaw University of Science and Technology, Wroclaw, Poland, 4 Department of Human Motor Skills, Faculty of Physical Education and Sport Science, Wroclaw University of Health and Sport Sciences, Wroclaw, Poland

☺ These authors contributed equally to this work.
* natalia.danek@awf.wroc.pl

**Data Availability Statement:** Data cannot be shared publicly because of concerns over the risk of inadvertent disclosure of personal health

## Abstract

The purpose of this study was to determine the effective warm-up protocol using an added respiratory dead space (ARDS) 1200 ml volume mask to determine hypercapnic conditions, on the swimming velocity of the 50 m time trial front crawl. Eight male members of the university swimming team, aged 19–25, performed three different warm-up protocols: 1) standardized warm-up in water (WU$_{CON}$); 2) hypercapnic warm-up in water (WU$_{ARDS}$); 3) hypercapnic a 20-minute transition phase on land, between warm-up in water and swimming test (RE-WU$_{ARDS}$). The three warm-up protocols were implemented in random order every 7th day. After each protocol, the 50 m time trial front crawl swimming (swimming test) was performed. The fastest time trial swimming of 50 m front crawl was achieved after the hypercapnic transition phase (RE-WU$_{ARDS}$) protocol and was 27.5 ± 1.6 seconds, 1.2% faster than hypercapnic warm-up protocol ($p = 0.01$). This result was confirmed by a higher swimming average speed of the exercise test after RE-WU$_{ARDS}$ compared to WU$_{ARDS}$ ($p = 0.01$). The use of ARDS provoked a state of tolerable hypercapnia (obtaining carbon dioxide concentration in arterialized blood $pCO_2 > 45$ mmHg) achieving a post-warm-up of WU$_{ARDS}$ value 49.7 ± 5.9 mmHg (compared to the control condition which was a statistically significant difference $p = 0.02$) and before time trial RE-WU$_{ARDS}$ 45.7 ± 2.1 mmHg ($p = 0.01$ compared to WU$_{CON}$). After breathing through the 1200 ml ARDS mask during the 20-minute re-warm-up phase, there was a trend of faster time trial among participants compared to the control condition, and statistically significantly faster times compared to WU$_{ARDS}$, indicating that further study is appropriate to verify the efficacy of the proposed method to improve swimming efficiency.

information and performance information of athletes. Data are available from the Wroclaw University of Health and Sport Sciences Research Ethics Committee (contact via e-m: zaklad.kz@awf. wroc.pl) for researchers who meet the criteria for access to confidential data. In order to proceed with permission to use the data set, it is necessary to conduct a joint investigation with the research staff.

**Funding:** The authors received no specific funding for this work.

**Competing interests:** All authors have declared that no competing interests exist.

## Introduction

Swimming efficiency depends on complex physiological, morphological, biomechanical, and motor factors [1]. At the level of the Olympic Games and World Championships, subtle improvements are crucial, as the time difference between a gold and silver medal is 0.01s (0.4% of the winner's time in the 50 m freestyle) [2]. Finding ways to improve performance by even 1% can have a significant impact on the final result.

Before training sessions and competitions, athletes perform warm-ups to maximize performance [3]. Warm-ups are aimed at increasing body temperature, lung ventilation, and heart rate (HR). Warm-ups are designed to increase blood flow through vasodilatation [4] and transport of oxygen to working muscles and faster deoxygenation of hemoglobin [3]. The warm-up process increases the initial oxygen uptake ($VO_2$) [5], improves nerve conduction speed [6] and reduces joint resistance [7]. This results in the optimization of metabolic reactions, particularly of the phosphagen and glycolytic systems, and thus improves muscle performance during dynamic exercise [8]. Various forms of a warm-up are commonly used, such as passive or active warm-ups, both in water or on land [3]. According to Maglischo [9], warm-ups performed in the water provides the "feel of the water" and increases the "readiness of the race".

The effect of the warm-up strategy on swimming performance also depends on the transition phase, which is the rest time between warm-up and start of the competitive event. A break longer than 20 minutes can rapidly reduce muscle temperature and impair the performance of the competition [10]. Various forms of counteracting the negative effects of the passive break are used, such as wearing warm clothing [11] or using exercises on land [12], also referred to as re-warm-up protocols [13]. Using active warm-up routines and combining them with the appropriate re-warm-up strategy can effectively improve swimming performance [12]. Ramos-Campo et al. [14] showed that active exercise on land under hypoxic conditions reduced the drop in body temperature during the transition phase, thereby improving the results of the 100 m time trial of young amateur swimmers. Provoking this condition by using specific devices in swimming pool is not easy and may be too expensive. In contrast, Robertson et al. [15] proposed an apnea series (temporary cessation of breathing) as a warm-up component to induce hypoxia. The response to the apnea series also induced hypercapnia, decreased blood saturation, increased acidosis, bradycardia, and splenic contraction due to, among other things, hypoxemia during the initial apneic phase [16]. This leads to an increase in the number of circulating erythrocytes, suggesting a potential method to rapidly increase the body's ability to transport oxygen. It should be mentioned that Robertson et al. [15] examined swimming performance during the 400m test, in which the aerobic energy system is dominant [17]. However, for improving anaerobic swimming performance (e.g. in the 50 m swimming sprint) other mechanisms should be stimulated during the re-warm-up phase.

One method to induce hypercapnia and respiratory acidosis is the use of added respiratory dead space (ARDS) [18]. Koopers et al. [19] used "tube breathing" (external dead space ventilation) and controlled breathing (modulating the rhythm of breathing) to improve the efficiency of respiratory muscles. The author found that in most participants tube breathing led to hypercapnia, which is well tolerated by healthy people. Breathing, through a specially constructed mask and tube, increases the retention of $CO_2$ during breathing and affects, through the chemoreceptive zones (central and peripheral), the work of the circulatory and respiratory systems, as well as the acid-base balance of the blood. The degree of these changes depends on the ARDS volume used and the intensity of effort [18]. It has been established that a capacity of 1200 ml during low-intensity exercise leads to hypercapnia [18]. Several studies have shown that this approach induces greater adaptive changes after a few weeks of regular training of swimmers [20, 21] and amateur triathletes [22]. Interesting results were shown in the study by

Danek et al. [23] using ARDS during warm-up before sprint interval exercise. This approach aimed "hypercapnic warm-up" induced ergogenic effects e.g. increasing temperature and bicarbonate ions concentration (improvement buffering capacity), which resulted in a statistically significantly higher mean power. However, the impact of a warm-up with 1200 ml ARDS in a single swimming sprint performance has not yet been verified.

The aim of this work is to analyze the impact of three warm-up protocols on the performance of swimming in a 50 m time trial using the front crawl technique. The three warm-up protocols included: 1) standardized warm-up in water ($WU_{CON}$); 2) warm-up with ARDS in water ($WU_{ARDS}$); 3) use of ARDS in the re-warm-up (a 20-minute transition phase on land), between a standardized warm-up and swimming test ($RE$-$WU_{ARDS}$). We hypothesized that both the combination of active warm-up in the water with the ARDS volume mask and the use of ARDS only in the transition phase as a re-warm-up would improve the performance of swimming the 50 m time trial front crawl compared to standard warm-up protocol. In addition, we hypothesized that the use of ARDS would not induce respiratory muscle fatigue under both experimental conditions.

## Materials and methods

### Participants

The study involved eight male participants with an average of 22.2 ± 2.4 years. All volunteers were students of physical education and were members of the university swimming team. All volunteers had the front crawl as their primary competitive stroke. The participants were swimmers with 12.9 ± 1.9 years of training experience. An interview with the head coach revealed that during the investigation which was conducted at the beginning of the winter general preparatory training period, the swimmers participated in dryland workouts and in-water practice. Swimmers trained 7–8 swimming (20–25 km) and 3 dryland sessions (2–3 hours) per week in the same squad and under the direction of the same coach. During the research period, the average activity of participants was limited to 18 ± 2.5 hours of training/week. To minimize any overtraining effects from the experiment, swimmers avoided stressful training during the days before the test. The sports level of the participants was determined by the best times in the 50 m freestyle competition (in the 25 m swimming pool), obtained in the last season amounting to 24.51 ± 1.37, categorized them as trained swimmers in their age group (565.8 ± 93.3 FINA points in a short course competition at the time of data collection).

The inclusion criteria were (1) at least 10 years of swimming training, (2) having the front crawl as their primary competition stroke, (3) participants were 26 years old and below (5) and able to give informed consent to participate. The exclusion criteria were (a) injuries three months before an experiment; (b) diagnosed asthma symptoms; and (c) smoking cigarettes. Similarities of the swimmers' somatic parameters (body mass, body height, fat mass, free fat mass) were used as an objective basis to compare their potential in terms of generated propulsive force quantified by the kinematic parameters i.e. stroke stride and stroke rate [24]. An objective assessment of the test group homogeneity was assessed with the Grubbs' Outlier Test [25] using the following formula (1): $g = max(x_i-\bar{x})·(SD)^{-1}$ where: (1) the maximal of the absolute differences between the values $x_i$ and the sample mean $\bar{x}$ divided by the standard deviation of the sample. If the resulting statistic test (g) is greater than the critical value ($g_{crit}$), the corresponding value can be regarded to be an outlier. An extract of the critical value for $n = 8$ (sample size) and at significance, level $p = 0.05$ is $g_{crit} = 2$. The results of the test found that the group was homogeneous in terms of somatic parameters and blood parameters (RBC, WBC, HGB). Participant characteristics and outcome of Grubbs' outlier test are presented in Table 1.

**Table 1. Participant characteristics (min, max, means (M), standard deviations (SD)) and outcome of Grubbs' outlier test.**

| Variables | Mean | SD | Min | Max | Grubbs' Outlier Test (g value) |
|---|---|---|---|---|---|
| Body mass (kg) | 76.6 | 5.7 | 66.1 | 84.5 | 1.96 |
| Body height (cm) | 180.1 | 7.6 | 170.0 | 192.0 | 1.66 |
| Fat mass (kg) | 10.4 | 1.9 | 7.3 | 12.7 | 1.73 |
| Free fat mass (kg) | 66.1 | 4.7 | 58.8 | 74.1 | 1.82 |
| RBC ($10^6 \cdot mm^{-3}$) | 4.78 | 0.2 | 4.5 | 5.0 | 1.74 |
| WBC ($10^3 \cdot mm^{-3}$) | 6.0 | 1.4 | 4.0 | 8.3 | 1.76 |
| HGB ($gl \cdot dL^{-1}$) | 14.6 | 0.7 | 13.6 | 15.3 | 1.69 |

Note: SD—standard deviation, g—Grubbs' Outlier Test outcome, *Grubbs' outlier test significant difference at $p < 0.05$ (for $g > g_{crit} = 2$); RBC—erythrocytes; WBC—leukocytes; HGB—hemoglobin concentration.

All participants were familiarized with the study procedure, both orally and in writing, and gave written, informed consent to participate. The participants were informed that they could withdraw from the experiment at any time. Participants were instructed to maintain a typical lifestyle during the experiment. The participants' diets were not controlled. All of the swimmers reported that they were free of drugs, medications, or dietary supplements known to influence physical performance. The study was approved by the local Research Ethics Committee (#1/2019) and carried out in accordance with the Helsinki Declaration in the Exercise Research Laboratory (certificate PN–EN ISO 9001:2009) and in a 25 m indoor swimming pool.

## Procedures

The investigation was carried out in January and February during the short-course season of the year. Since none of the participants had previously used the ARDS mask, a session in the swimming pool was conducted in the first week to familiarize the participants with this method. The familiarization session included breathing on land through ARDS and swimming 1000 m front crawl on low-intensity in a 25 m indoor pool (water temperature 27˚C, air temperature 28˚C, relative humidity 60%, lighting 600 lux). When breathing with ARDS, a special nose plug was used to eliminate nasal breathing. The ARDS apparatus was constructed using a swimming snorkel with a mouthpiece (Speedo International Ltd., Nottingham, UK) connected to a ribbed tube with a diameter of 2.5 cm, in order to increase the anatomical dead space by 1200 ml (Fig 1). ARDS was identical for each participant and measured by filling a pipe with water and then transferring the volume to a measuring cylinder [20]. The ARDS was stiff enough to maintain a constant volume while swimming. During previous studies, after swimming with the ARDS apparatus, participants were individually interviewed to ask if there was any discomfort in wearing a snorkel [20, 21, 26]. The majority of participants said that there wasn't any discomfort in wearing the apparatus. The ARDS mask was used only in warm-up techniques in water or on land; it was not used in the 50 m time trial front crawl swimming performance.

During subsequent visits in the swimming pool, the participants were exposed to three separate warm-up (WU) protocols, in random order, every seventh day (Fig 2):

$WU_{CON}$—conventional warm-up in water as a control condition with a 20-minute transition phase (while seated on the pool deck in a control condition) between warm-up and swimming test.

$WU_{ARDS}$—warm-up in water with ARDS with a 20-minute transition phase (while seated on the pool deck in a control condition) between warm-up and swimming test in a control condition.

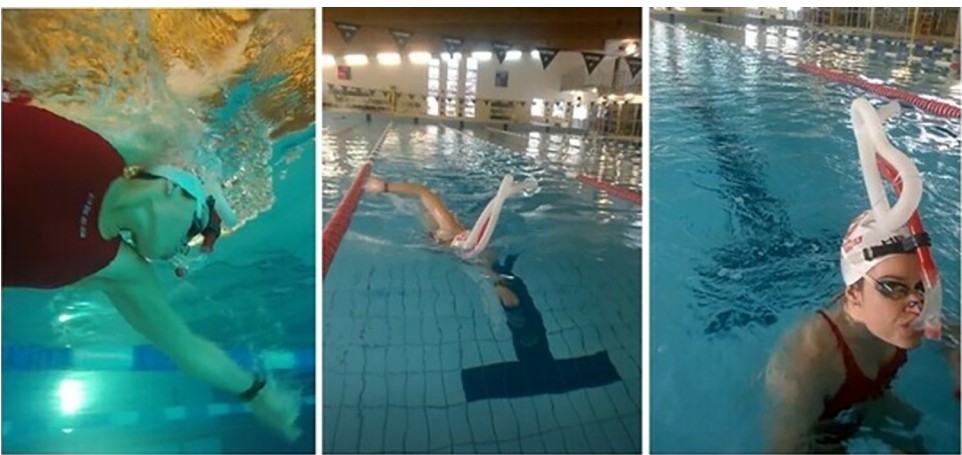

**Fig 1. The instrument increasing added respiratory dead space: A custom added respiratory dead space (ARDS) apparatus consisting of a polypropylene center-mount swimming snorkel with a mouthpiece (Speedo International Ltd.,. Nottingham, UK) integrated with 2.5-cm diameter ribbed tubing to provide 1200 ml of dead space.**

RE-WU$_{ARDS}$—conventional warm-up in water with a 20-minute transition phase (while seated on the pool deck in a control condition with application of ARDS) between warm-up and swimming test.

After each protocol, the 50 m time trial front crawl swimming was performed.

The conventional warm-up was designed based on the starting routine of the participants and strategy by Dalamitros et al. [27]. The warm-up covered a total distance of 1000 m and consisted of the following parts: 300 m swim (easy pace); 6 × 50 m swim starting at 1:15 (pull/kick/drill); 8 × 25 m starting at 1:00 (4 × 25 m: 12.5 m 90% of the 50 m race pace followed by 12.5 m easy and 4 × 25 m vice versa); 2 × 50 m starting at 2:00 (25 m at 100% of the 50 m race pace followed by 25 m at an easy pace); 100 m easy swim. During WU$_{ARDS}$, participants breathed through the 1200 ml ARDS apparatus (Fig 1).

After warming up in the water, there was passive restitution on land, which lasted 20 minutes. At that time, the participants sat at rest on a chair. Participants were covered with a towel to prevent excessive loss of body heat. During the RE-WU$_{ARDS}$, participants breathed with an ARDS apparatus while at rest (Fig 1).

Immediately after the passive restitution, a 50 m time trial was conducted, consisting of swimming at maximal speed using the front crawl technique without the ARDS mask. The test was started from a stationary position lying on the chest in the water. The measurements were

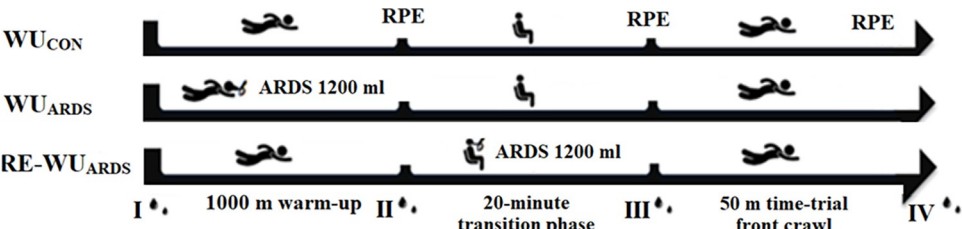

**Fig 2. Protocols used in each test procedure with measurement time points.** I—resting, II—after warm-up, III—before time-trial, IV—after time-trial. WU$_{CON}$—warm-up in water, WU$_{ARDS}$—warm-up in water with ARDS, RE-WU$_{ARDS}$—warm-up in water with application of ARDS on land during the transition phase between warm-up and swimming test. RPE—Borg scale.

taken by a specialized lab technician with a device calibrated before each trial. During data collection, the participants stayed in the first three lanes of the pool, to which others had no access.

Before each time trial, participants were asked to refrain from strenuous exercise (24 hours), alcohol (24 hours), caffeine (12 hours), and food (2 hours) [28].

### Anthropometric measurements

During the tests in the laboratory, the weight and height of each participant were measured on the WPT 200 medical scale (Radwag, Radom, Poland). Body mass index (BMI; $kg \cdot m^{-2}$), fat mass (FM; kg), and fat-free mass (FFM; kg) were determined by near-infrared interactance (Futrex Tech, Inc., Gaithersburg, USA) at the middle of the biceps brachii muscle of the dominant extremity. Measurements were collected pre- and post-intervention as an indirect measure of changes in lipid metabolism. Measurements were taken by a laboratory member with the device calibrated before each trial [29].

### Methods for assessing blood parameters

Arterialized blood was collected from the fingertip to determine blood counts: erythrocytes (RBC $10^6 \cdot mm^{-3}$), leukocytes (WBC $10^3 \cdot mm^{-3}$), and hemoglobin concentration (HGB $gl \cdot dL^{-1}$) using the ABX Micros OT.16 device (Horiba Medical, Kyoto Japan). Blood parameters were measured to assess the hemodynamic characteristics of the participants.

### Methods for the assessment of physiological parameters

Capillary blood samples were taken four times from the fingertip immediately before the initial phase of warm-up, immediately after the warm-up, 2 minutes before the 50 m time trial, and 3 minutes after the time trial. The procedure was performed to determine the acid-base balance of blood using Rapid Lab 348 (Bayer, Leverkusen, Germany): pH ($-logH^+$), $pCO_2$ (mmHg), $pO_2$ (mmHg), and lactate $La^-$ ($mmoL \cdot L^{-1}$) with Dr. Lange 140 photometer (LP 400 Dr Lange, Berlin, Germany). The concentration of hydrogen ions ($H^+$) was calculated based on the blood pH scale using the formula $(H+) = 10^{-pH}$.

During swimming, heart rate (HR) was monitored using the V-800 sport tester (Polar Electro, Finland). The peak heart rate (HRpeak) value ($b \cdot min^{-1}$) was determined from the digital HR record.

### Respiratory muscle strength variables measurements

The strength of inspiratory muscles PImax ($cmH_2O$) (inspirers maximal pressure) and expiratory muscles PEmax ($cmH_2O$) (expiratory maximal pressure) was measured using the Micro RPM device—Respiratory Pressure Meter (CareFusion, San Diego, USA). In the standing position, the participant took the maximal inhalation from the maximal exhalation level in order to measure PImax. From the maximal inhalation, the participant exhaled to measure PEmax. In both cases, a special nose plug was used. The test was conducted at rest before the start of the warm-up ($PImax_I$, $PEmax_I$) and immediately following the 50 m time trial ($PImax_{IV}$, $PEmax_{IV}$). Each participant performed two tests (max inhalation and max exhalation), from which the higher value was selected for further analysis.

### Rating of perceived exertion measure

Subjective assessment of perceived exertion was measured using the Borg scale (RPE—rate of perceived exertion). The Borg scale consists of 15 levels (6–20) with a rating of 6 indicating no

exertion and a rating of 20 indicating maximal exertion [30]. The participants assessed their perceived effort in the water immediately after each time test. For further statistical analysis, the indicated scale values were used.

## Methods for the assessment of swimming kinematics during a swimming test

**Time measure.** The time (t) of covering the distance of 50 m was measured using a Finis 3X300M stopwatch (Finis, Tracy, USA). The first 25 m ($t_{25(1)}$) and the second 25 m ($t_{25(2)}$) during the entire 50 m distance were also evaluated. The measurement was made by two researchers, and the average time of both measurements was used for the analysis.

**Stroke rate measure.** Stroke rate (SR) is the number of complete arm cycles executed in a given time (e.g. minute). The assessment of the swimmer's movements took place in the measurement window 10 m from the wall when the swimmer completed one full arm cycle after break of the surface water (breakout phase). On the track line, a marker indicated the distance of 10 m from the wall. At the 50 m time trial, the measurement was made twice on the first and second sections of 25 m. During the 50 m time trial, the experimenter measured execution of an arm cycles by a swimmer by Finis Pace Clock 3X300M stopwatch (Finis, Tracy USA). The stopwatch was started on the entry of the right hand and stopped on the entry of the same hand after executing three cycles [31]. SR was calculated using the following formula [32] (1): $SR = 60 \cdot (3 \cdot t3c^{-1})$ $(cycles \cdot min^{-1})$ (1), where: SR—stroke rate, 3—number of measured cycles, t3c - time of 3 cycles.

**Stroke length measure.** Stroke length (SL) is the horizontal distance traveled during the completion of one complete cycle of the swimmer's arms. SL was calculated for the first 25 m and the entire test distance of 50 m. The SR formula (1) was converted to calculate the SL [32] (2): $SL = v \cdot (60 \cdot SR^{-1})$ $(m \cdot cycle^{-1})$ (2), where: SL—stroke length, v—average swimming speed, SR —stroke rate.

**Swimming average speed measure.** The average horizontal swimming speed (v) was calculated for the first 25 m and the entire test distance of 50 m. The swimming speed was calculated using the following formula (3): $v = d \cdot t^{-1}$ $(m \cdot s^{-1})$ (3), where: v—average swimming speed, d—length in meters of the distance covered, t—time in seconds of distance covered.

**Stroke index.** Stroke index (SI) was determined for the first 25 m and the entire test distance of 50 m as the product of average velocity and stroke length. SI was calculated using the following formula [33] (4): $SI = SL \cdot v$ $(m^2 \cdot (s \cdot cycle)^{-1})$ (4), where: SI—stroke index, SL—stroke length, v—average swimming speed.

## Statistical analyses

Means, standard deviation, and confidence intervals were calculated for all variables. Significance was set at an alpha level of $< 0.05$ for all statistical procedures, with $p$ values provided for all results. The Shapiro-Wilk test was used to assess the differences between the distribution of the sample and the theoretical normal distribution. Using Mauchly's test, the variance was checked for differences between individual measurements. The assumption of normal distribution and sphericity was fulfilled.

The quantitative study analysis involved a 4-dimensional approach (alpha, power, sample size, and effect size) [34] and was computed using G*Power 3.1.9.2 software (Franz Faul, Kiel, Germany) [35]. The sample size of $n = 8$ was set with a minimum acceptable effect size of $f^2 = 0.02$ [36], with the level of significance set as a = 0.05, and a power of $1-\beta = 0.05$. Effect sizes (partial eta squared ($\eta^2$)) for ANOVA were interpreted as small (0.02), moderate (0.13), or large ($\geq 0.26$) [36].

The differences between each of the analyzed dependent variables between the three protocols of warm-up test protocols were assessed by a parametric test of univariate analysis of variance ANOVA in a single-variate model. Detailed pairs comparisons were made using a correction for multiple comparisons—Bonferroni's post hoc test. Effect sizes for ANOVA were calculated by using partial eta squared ($\eta^2$). Effect sizes were interpreted as small (0.02), moderate (0.13), or large ($\geq 0.26$) [36].

Differences in physiological parameters and parameters of the swimming cycle obtained in the $WU_{ARDS}$ protocol and $RE\text{-}WU_{ARDS}$ protocol in relation to $WU_{CON}$ were used to search for dependencies in relation to time trial $t_{50}$ and splits $t_{25(1)}$, $t_{25(2)}$. Pearson's correlation coefficient (r) was used to describe the strength of the correlation. A correlation was interpreted as none: 0; poor: 0.1–0.2; fair: 0.3 to 0.5; moderate: 0.6–07; very strong: 0.8–0.9; perfect: 1 [37]. Differences tests and correlations analysis were performed with the IBM SPSS Statistics version 26 software package (IBM, Inc., Chicago, USA).

Multiple linear regression (MLR) ($R^2$) with ordinary least squares (OLS) method [38] was used to show relationships between following dependent (explained) variables ($y$) time trial time $t_{50}$, $t_{25(1)}$, $t_{25(2)}$, and differences in selected independent variables (explanatory) ($x$) $La^-$, $HCO_3^-$, $H^+$, RPE ($_{II}$—after warm-up, $_{III}$—before time trial, $_{IV}$—after time trial) obtained in the $WU_{ARDS}$ and $RE\text{-}WU_{ARDS}$ protocols, against $WU_{CON}$. The determination coefficient was interpreted as unsatisfactory: 0.0–0.5; poor: 0.5–0.6; satisfactory: 0.6–0.8; good: 0.8–0.9; very good: 0.9–1.0. The numerical value also explains the percentage of standard deviation [39]. Before performing regression analysis, the Durbin-Watson value was identified for autocorrelation was also calculated with assuming an acceptable range of 1.50–2.50 [40]. Two indicators obtained from the linear regression analysis, namely variance inflation factors (VIF) and tolerance (TOL), were employed to detect the potential multicollinearity problem. The VIF $> 4$ or TOL $< 0.25$ indicates a multicollinearity problem [41]. Regression analysis was performed with the high-level general-purpose programming language Python 3.8.5 (Python Software Foundation, Wilmington, USA).

## Results

Among the physiological variables, before the time trial ($pCO_{2\ III}$), a statistically significant higher $pCO_2$ ($p = 0.00$) was observed in the main effect, as well as significant interactions in the $RE\text{-}WU_{ARDS}$ warm-up protocol compared to $WU_{CON}$ ($p = 0.00$) and $WU_{ARDS}$ ($p = 0.01$) protocols.

Before the time trial (III), the main effect of pH showed statistical significance at $p = 0.03$, while no significant changes were observed in the interaction between $WU_{CON}$ and $WU_{ARDS}$ ($p > 0.05$). The interaction between $WU_{ARDS}$ and $RE\text{-}WU_{ARDS}$ showed a lower pH in $RE\text{-}WU_{ARDS}$ compared to $WU_{ARDS}$ ($p = 0.03$) (Table 2). Despite statistical significance in the main effect of $HCO_3^-{}_I$ ($p = 0.04$), no significant interaction between conditions was observed. Similar results were observed in the analysis of $La^-$ changes. Only statistical significance was found in the main effect when analyzing $La^-$ changes after time trial ($La^-{}_{IV}$).

Among the kinematic movement variables, no statistically significant changes in the main effect were observed (Table 3). Despite the lack of significance and the observed differences between the means, which were not detected by ANOVA, a post hoc analysis was performed to verify the interaction between conditions, based on the study of Midway et al. [42]. As a result of this analysis, a statistically significant shorter time trial was observed following the $RE\text{-}WU_{ARDS}$ protocol compared to $WU_{ARDS}$ ($p = 0.01$). This result was confirmed by a higher average swim speed in the time trial after $RE\text{-}WU_{ARDS}$ compared to $WU_{ARDS}$ ($p = 0.01$) (Fig 3).

**Table 2. Analysis of physiological parameters for different warm-up test protocols.**

| Variable | Protocol | Mean | SD | ANOVA 95% confidence interval | | |
|---|---|---|---|---|---|---|
| | | | | F | p | $\eta^2$ |
| $Po_{2\ I}$ (mmHg) | $WU_{CON}$ | 66.99 | 7.99 | 0.23 | 0.65 | 0.03 |
| | $WU_{ARDS}$ | 65.36 | 5.01 | | | |
| | $RE\text{-}WU_{ARDS}$ | 65.53 | 3.04 | | | |
| $pO_{2\ II}$ (mmHg) | $WU_{CON}$ | 80.76 | 7.19 | 5.09 | 0.06 | 0.42 |
| | $WU_{ARDS}$ | 78.95 | 10.05 | | | |
| | $RE\text{-}WU_{ARDS}$ | 86.06 | 3.792 | | | |
| $pO_{2\ III}$ (mmHg) | $WU_{CON}$ | 65.43 | 8.28 | 0.07 | 0.79 | 0.01 |
| | $WU_{ARDS}$ | 63.44 | 3.06 | | | |
| | $RE\text{-}WU_{ARDS}$ | 64.61 | 5.49 | | | |
| $pO_{2\ IV}$ (mmHg) | $WU_{CON}$ | 77.20 | 14.43 | 3.01 | 0.13 | 0.30 |
| | $WU_{ARDS}$ | 76.24 | 14.08 | | | |
| | $RE\text{-}WU_{ARDS}$ | 71.04 | 10.47 | | | |
| $pCO_{2\ I}$ (mmHg) | $WU_{CON}$ | 41.74 | 2.41 | 5.58 | 0.05 | 0.44 |
| | $WU_{ARDS}$ | 42.45 | 2.56 | | | |
| | $RE\text{-}WU_{ARDS}$ | 43.85 | 2.18 | | | |
| $pCO_{2\ II}$ (mmHg) | $WU_{CON}$ | 41.34 | 2.61 | 2.26 | 0.18 | 0.24 |
| | $WU_{ARDS}$ | 49.74 | 5.90 | | | |
| | $RE\text{-}WU_{ARDS}$ | 42.80 | 1.07 | | | |
| $pCO_{2\ III}$ (mmHg) | $WU_{CON}$ | 41.86 | 1.93 | 133.25 | **0.00*** | 0.95 |
| | $WU_{ARDS}$ | 42.28 | 3.30 | | | |
| | $RE\text{-}WU_{ARDS}$ | 45.65 | 2.11 | | | |
| $pCO_{2\ IV}$ (mmHg) | $WU_{CON}$ | 40.18 | 3.58 | 3.62 | 0.10 | 0.34 |
| | $WU_{ARDS}$ | 41.50 | 4.20 | | | |
| | $RE\text{-}WU_{ARDS}$ | 44.23 | 4.02 | | | |
| $pH_I$ | $WU_{CON}$ | 7.37 | 0.03 | 0.01 | 0.92 | 0.00 |
| | $WU_{ARDS}$ | 7.38 | 0.02 | | | |
| | $RE\text{-}WU_{ARDS}$ | 7.37 | 0.03 | | | |
| $pH_{II}$ | $WU_{CON}$ | 7.30 | 0.05 | 0.20 | 0.67 | 0.03 |
| | $WU_{ARDS}$ | 7.27 | 0.05 | | | |
| | $RE\text{-}WU_{ARDS}$ | 7.30 | 0.04 | | | |
| $pH_{III}$ | $WU_{CON}$ | 7.37 | 0.02 | 7.70 | **0.03*** | 0.52 |
| | $WU_{ARDS}$ | 7.37 | 0.03 | | | |
| | $RE\text{-}WU_{ARDS}$ | 7.34 | 0.03 | | | |
| $pH_{IV}$ | $WU_{CON}$ | 7.22 | 0.06 | 0.74 | 0.42 | 0.10 |
| | $WU_{ARDS}$ | 7.21 | 0.05 | | | |
| | $RE\text{-}WU_{ARDS}$ | 7.23 | 0.05 | | | |
| $HCO_3^-{}_{\ I}$ (mmoL·$L^{-1}$) | $WU_{CON}$ | 24.03 | 0.81 | 6.35 | **0.04*** | 0.48 |
| | $WU_{ARDS}$ | 24.68 | 0.82 | | | |
| | $RE\text{-}WU_{ARDS}$ | 25.03 | 1.16 | | | |
| $HCO_3^-{}_{\ II}$ (mmoL·$L^{-1}$) | $WU_{CON}$ | 20.16 | 2.44 | 0.48 | 0.51 | 0.07 |
| | $WU_{ARDS}$ | 22.50 | 2.39 | | | |
| | $RE\text{-}WU_{ARDS}$ | 20.63 | 2.25 | | | |
| $HCO_3^-{}_{\ III}$ (mmoL·$L^{-1}$) | $WU_{CON}$ | 23.45 | 1.26 | 0.65 | 0.45 | 0.09 |
| | $WU_{ARDS}$ | 23.68 | 2.37 | | | |
| | $RE\text{-}WU_{ARDS}$ | 23.88 | 0.77 | | | |

(*Continued*)

**Table 2.** (Continued)

| Variable | Protocol | Mean | SD | ANOVA 95% confidence interval | | |
|---|---|---|---|---|---|---|
| | | | | F | p | $\eta^2$ |
| $HCO_3^-{}_{IV}$ (mmoL· L$^{-1}$) | $WU_{CON}$ | 16.10 | 2.85 | 5.09 | 0.06 | 0.42 |
| | $WU_{ARDS}$ | 16.50 | 3.13 | | | |
| | $RE\text{-}WU_{ARDS}$ | 18.15 | 2.51 | | | |
| $La^-{}_I$ (mmoL· L$^{-1}$) | $WU_{CON}$ | 1.60 | 0.32 | 0.37 | 0.56 | 0.05 |
| | $WU_{ARDS}$ | 1.50 | 0.24 | | | |
| | $RE\text{-}WU_{ARDS}$ | 1.69 | 0.20 | | | |
| $La^-{}_{II}$ (mmoL· L$^{-1}$) | $WU_{CON}$ | 4.70 | 2.93 | 0.43 | 0.53 | 0.06 |
| | $WU_{ARDS}$ | 3.75 | 1.82 | | | |
| | $RE\text{-}WU_{ARDS}$ | 4.30 | 1.82 | | | |
| $La^-{}_{III}$ (mmoL· L$^{-1}$) | $WU_{CON}$ | 2.75 | 1.69 | 0.36 | 0.57 | 0.05 |
| | $WU_{ARDS}$ | 2.24 | 0.68 | | | |
| | $RE\text{-}WU_{ARDS}$ | 2.43 | 0.55 | | | |
| $La^-{}_{IV}$ (mmoL· L$^{-1}$) | $WU_{CON}$ | 9.83 | 2.64 | 6.27 | **0.04*** | 0.47 |
| | $WU_{ARDS}$ | 9.09 | 2.67 | | | |
| | $RE\text{-}WU_{ARDS}$ | 7.98 | 1.81 | | | |
| $RPE_{II}$ | $WU_{CON}$ | 9.50 | 1.41 | 0.00 | 1.00 | 0.00 |
| | $WU_{ARDS}$ | 10.63 | 2.67 | | | |
| | $RE\text{-}WU_{ARDS}$ | 9.50 | 1.60 | | | |
| $RPE_{III}$ | $WU_{CON}$ | 6.13 | 0.35 | 3.40 | 0.11 | 0.33 |
| | $WU_{ARDS}$ | 6.88 | 1.13 | | | |
| | $RE\text{-}WU_{ARDS}$ | 7.25 | 1.75 | | | |
| $RPE_{IV}$ | $WU_{CON}$ | 14.50 | 2.07 | 3.10 | 0.12 | 0.31 |
| | $WU_{ARDS}$ | 15.25 | 1.75 | | | |
| | $RE\text{-}WU_{ARDS}$ | 15.63 | 1.69 | | | |

Note: Data presented as mean ± standard deviation. *Significant difference at $p < 0.05$. $_I$—resting, $_{II}$—after warm-up, $_{III}$—before time trial, $_{IV}$—after time trial. $WU_{CON}$—warm-up in water, $WU_{ARDS}$—warm-up in water with ARDS, $RE\text{-}WU_{ARDS}$—warm-up in water with application of ARDS on land during the transition phase between warm-up and swimming test; pH—acid-base balance; $pCO_2$ -partial pressure of carbon dioxide in arterialized blood, $pO_2$—partial pressure of oxygen in arterialized blood, $HCO_3^-$ - bicarbonate concentration in arterialized blood, $La^-$—lactate concentration, RPE–rate of perceived exertion (Borg's scale).

No differences were found among the respiratory muscle strength variables (S1 Table).

Among the physiological parameters in the $WU_{ARDS}$ protocol, the concentration of $pCO_2{}_{II}$ (after warm-up) significantly correlates with the first intertime $t_{25(1)}$ ($p = 0.04$). In the $RE\text{-}WU_{ARDS}$ protocol, the concentration of $HCO_3^-{}_I$ (resting) significantly correlates with the first intertime $t_{25(1)}$ rest ($p = 0.02$) and the time trial time $t_{50}$ rest ($p = 0.02$). In addition, the concentration of $La^-{}_{IV}$ (after time trial) significantly correlates with the second time $t_{25(2)}$ rest ($p = 0.03$) and the time trial time $t_{50}$ rest ($p = 0.03$) (Table 4).

Among the kinematic parameters of motion in the $RE\text{-}WU_{ARDS}$ protocol, the variable $v_{25(1)}$ significantly correlates with the second intertime $t_{25(2)}$ rest ($p = 0.04$) and the variable $v_{25(2)}$ significantly correlates with the second intertime $t_{25(2)}$ rest ($p = 0.01$) (Table 4).

Among the kinematic parameters of the swimming cycle in the $RE\text{-}WU_{ARDS}$ protocol, significant correlations with the time trial $t_{50}$ rest test time were recorded for the variables $SR_{25(1)}$ ($p = 0.04$), $SL_{25(1)}$ ($p = 0.04$), $SI_{25(1)}$ ($p = 0.04$) (Table 5). No other significant correlations were observed in the assessed parameters.

Multiple linear regression model ($R^2 = 0.93$) was recorded between the time $t_{25(2)}$ and the differences in the predictor variables $La^-$ ($p = 0.03$), $HCO_3^-$ ($p = 0.02$), $H^+$ ($p = 0.046$) obtained

**Table 3. Analysis of kinematic parameters of movement and swimming cycle during a 50 m front crawl (time trial front crawl) for various warm-up test protocols.**

| Variable | Protocol | Mean | SD | ANOVA 95% confidence interval | | |
|---|---|---|---|---|---|---|
| | | | | F | p | η2 |
| $t_{25(1)}$ (s) | $WU_{CON}$ | 13.79 | 1.00 | 0.11 | 0.75 | 0.02 |
| | $WU_{ARDS}$ | 13.74 | 0.92 | | | |
| | $RE\text{-}WU_{ARDS}$ | 13.88 | 0.79 | | | |
| $t_{25(2)}$ (s) | $WU_{CON}$ | 14.02 | 0.77 | 1.81 | 0.22 | 0.21 |
| | $WU_{ARDS}$ | 14.10 | 0.97 | | | |
| | $RE\text{-}WU_{ARDS}$ | 13.63 | 1.28 | | | |
| $t_{50}$ (s) | $WU_{CON}$ | 27.81 | 1.75 | 1.25 | 0.30 | 0.15 |
| | $WU_{ARDS}$ | 27.84 | 1.73 | | | |
| | $RE\text{-}WU_{ARDS}$ | 27.51 | 1.64 | | | |
| $V_{25(1)}$ (m·s$^{-1}$) | $WU_{CON}$ | 1.82 | 0.13 | 0.16 | 0.70 | 0.02 |
| | $WU_{ARDS}$ | 1.83 | 0.13 | | | |
| | $RE\text{-}WU_{ARDS}$ | 1.81 | 0.11 | | | |
| $V_{25(2)}$ (m·s$^{-1}$) | $WU_{CON}$ | 1.79 | 0.10 | 2.06 | 0.20 | 0.23 |
| | $WU_{ARDS}$ | 1.78 | 0.12 | | | |
| | $RE\text{-}WU_{ARDS}$ | 1.85 | 0.19 | | | |
| $V_{50}$ (m·s$^{-1}$) | $WU_{CON}$ | 1.80 | 0.12 | 1.32 | 0.29 | 0.16 |
| | $WU_{ARDS}$ | 1.80 | 0.11 | | | |
| | $RE\text{-}WU_{ARDS}$ | 1.82 | 0.11 | | | |
| $SR_{25(1)}$ (cycles· min$^{-1}$) | $WU_{CON}$ | 66.19 | 7.57 | 2.97 | 0.13 | 0.30 |
| | $WU_{ARDS}$ | 70.37 | 11.94 | | | |
| | $RE\text{-}WU_{ARDS}$ | 72.78 | 10.37 | | | |
| $SR_{25(2)}$ (cycles· min$^{-1}$) | $WU_{CON}$ | 66.96 | 7.64 | 0.01 | 0.91 | 0.00 |
| | $WU_{ARDS}$ | 65.27 | 8.50 | | | |
| | $RE\text{-}WU_{ARDS}$ | 66.48 | 8.48 | | | |
| $SL_{25(1)}$ (m·cycle$^{-1}$) | $WU_{CON}$ | 1.67 | 0.21 | 2.59 | 0.15 | 0.27 |
| | $WU_{ARDS}$ | 1.61 | 0.35 | | | |
| | $RE\text{-}WU_{ARDS}$ | 1.51 | 0.18 | | | |
| $SL_{25(2)}$ (m·cycle$^{-1}$) | $WU_{CON}$ | 1.62 | 0.23 | 0.45 | 0.53 | 0.06 |
| | $WU_{ARDS}$ | 1.67 | 0.30 | | | |
| | $RE\text{-}WU_{ARDS}$ | 1.69 | 0.27 | | | |
| $SI_{25(1)}$ (m$^2$·(s·cycle)$^{-1}$) | $WU_{CON}$ | 3.05 | 0.55 | 2.08 | 0.19 | 0.23 |
| | $WU_{ARDS}$ | 2.97 | 0.81 | | | |
| | $RE\text{-}WU_{ARDS}$ | 2.72 | 0.31 | | | |
| $SI_{25(2)}$ (m$^2$·(s·cycle)$^{-1}$) | $WU_{CON}$ | 2.92 | 0.56 | 1.00 | 0.35 | 0.13 |
| | $WU_{ARDS}$ | 3.00 | 0.71 | | | |
| | $RE\text{-}WU_{ARDS}$ | 3.16 | 0.78 | | | |

Note: Data presented as mean ± standard deviation. *Significant difference at $p < 0.05$. t–time of trial, V—swimming speed, SR—stroke rate, SL—stroke length, SI—stroke index, $_{25(1)}$—first 25 m, $_{25(2)}$—second 25 m; $_{50}$—the whole distance. $WU_{CON}$—warm-up in water, $WU_{ARDS}$—warm-up in water with ARDS, RE-WUARDS—warm-up in water with application of ARDS on land during the transition phase between warm-up and swimming test.

in the $WU_{ARDS}$ protocol, relative to the control condition ($WU_{CON}$) in the measurement before time trial $_{(III)}$, meaning that the model explained 93% of the variance. Durbin-Watson (*DW*) value was identified with value $DW = 1.64$. Since the Durbin-Watson coefficient is in the normal range (1.5–2.5), meaning that the study meets the regression assumptions. According to Field [43], the closer the Durbin-Watson coefficient is to 2, the stronger the assumption.

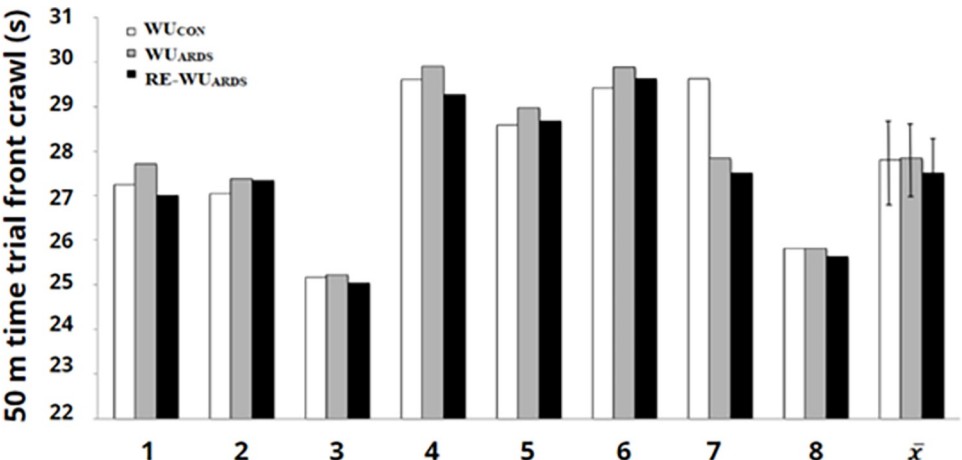

**Fig 3. Individually obtained time during the time trial 50 m front crawl (1–8) and the average group ($\bar{x}$) and bars of errors describing the standard deviation (SD) in various warm-up test procedures.**

VIF and TOL values for predictor variables in the model (respectively: 1.02–1.24, 0.81–0.99) showed there is no multicollinearity [44].

## Discussion

This research compared the effect of three different warm-up protocols on the swimmers 50 m time trial front crawl sprint. Although previous studies have tested the intervention of ARDS in swimming as a strategy to improve cardiorespiratory function [20] or lung and respiratory muscle function [21], to our knowledge, this is the first study to examine the effects of warming up and re-warm up with ARDS in university-level swimmers. The results indicate that 20 minutes of breathing through a 1200 ml added respiratory dead space volume mask during re-warm-up led to a reduction in the 50 m time trial. Moreover, we confirmed the hypothesis that breathing through ARDS with a diameter of 2.5 cm does not lead to respiratory muscle fatigue, therefore it does not adversely affect the exercise capacity of swimmers.

### Hypercapnic conditions and lactate concentration

In both warm-up protocols using ARDS, hypercapnia occurred. According to Patz et al. [45], breathing through an additional volume of respiratory dead space increases $FiCO_2$ (the $CO_2$ fraction in inspiratory air), which was confirmed by Danek et al. [23]. Hence, it is not surprising that blood pH was lower when breathing by ARDS; evidence of the occurrence of respiratory acidosis (hypercapnic), is also reported by Smołka et al. [46]. Respiratory acidosis affects the acid-base balance of the muscle system, which can induce metabolic changes. Although the lactate concentration after the 50 m test did not differ among the three warm-up protocols used, there is a trend towards lower values after RE-WU$_{ARDS}$. Similar blood lactate concentrations ($9.1 \pm 1.9$ mmoL·$L^{-1}$) after 50 m front crawl swimming in national-level athletes were reported by Vescovi et al. [47]. Several studies have found that increased $CO_2$ partial pressure in the blood can lead to lower lactate release from muscles [48, 49]. In the previous work published by Danek et al. [23], after a 15-minute exercise at an intensity of 60% VO$_2$max, a decrease in lactate concentration, despite significantly lower blood pH and higher pCO$_2$, was observed. On the other hand, Smołka et al. [46] found no differences in La$^-$ when participants breathed through 1200 ml of ARDS while performing 30 minutes of exercise at an intensity of 60%VO$_2$max. We cannot completely rule out that ARDS breathing led to less lactate

**Table 4. Results of the Pearson correlation (r) between time trial $t_{50}$ and between times $t_{25(1)}$, $t_{25(2)}$ and the difference Δ physiological parameters obtained in the $WU_{ARDS}$ protocol and $RE\text{-}WU_{ARDS}$, relative to $WU_{CON}$.**

| Variable | | $\Delta WU_{ARDS}$—$WU_{CON}$ | | | $\Delta RE\text{-}WU_{ARDS}$—$WU_{CON}$ | | |
|---|---|---|---|---|---|---|---|
| | | $t_{25(1)}$ | $t_{25(2)}$ | $t_{50}$ | $t_{25(1)}$ | $t_{25(2)}$ | $t_{50}$ |
| $pCO_2$ I mmHg) | r | 0.11 | -0.06 | 0.03 | 0.62 | -0.04 | 0.27 |
| | p | 0.79 | 0.89 | 0.95 | 0.11 | 0.93 | 0.52 |
| $pCO_2$ II (mmHg) | r | **-0.74** | -0.51 | -0.69 | -0.18 | -0.48 | -0.46 |
| | p | **0.04***  | 0.19 | 0.06 | 0.67 | 0.22 | 0.25 |
| $pCO_2$ III (mmHg) | r | -0.03 | 0.12 | 0.05 | 0.20 | -0.33 | -0.16 |
| | p | 0.94 | 0.79 | 0.91 | 0.64 | 0.43 | 0.71 |
| $pCO_2$ IV (mmHg) | r | -0.07 | -0.12 | -0.11 | -0.13 | -0.56 | -0.50 |
| | p | 0.86 | 0.77 | 0.80 | 0.76 | 0.15 | 0.21 |
| pH I | r | 0.28 | 0.10 | 0.21 | -0.02 | 0.28 | 0.21 |
| | p | 0.50 | 0.82 | 0.63 | 0.96 | 0.50 | 0.62 |
| pH II | r | 0.40 | 0.27 | 0.37 | 0.27 | -0.03 | 0.11 |
| | p | 0.32 | 0.52 | 0.37 | 0.51 | 0.94 | 0.80 |
| pH III | r | 0.21 | 0.15 | 0.20 | 0.14 | 0.24 | 0.25 |
| | p | 0.62 | 0.72 | 0.64 | 0.75 | 0.58 | 0.55 |
| pH IV | r | -0.42 | -0.37 | -0.43 | -0.24 | -0.23 | -0.29 |
| | p | 0.31 | 0.37 | 0.29 | 0.56 | 0.59 | 0.48 |
| $HCO_3^-$ I (mmoL·$L^{-1}$) | r | 0.66 | 0.19 | 0.46 | **0.78** | 0.52 | **0.78** |
| | p | 0.07 | 0.66 | 0.25 | **0.02***  | 0.19 | **0.02***  |
| $HCO_3^-$ II (mmoL·$L^{-1}$) | r | -0.55 | -0.36 | -0.50 | 0.07 | -0.38 | -0.26 |
| | p | 0.16 | 0.38 | 0.21 | 0.86 | 0.35 | 0.53 |
| $HCO_3^-$ III (mmoL·$L^{-1}$) | r | 0.13 | 0.15 | 0.16 | 0.22 | 0.14 | 0.21 |
| | p | 0.75 | 0.72 | 0.71 | 0.61 | 0.74 | 0.61 |
| $HCO_3^-$ IV (mmoL·$L^{-1}$) | r | -0.43 | -0.34 | -0.42 | -0.26 | -0.52 | -0.53 |
| | p | 0.29 | 0.41 | 0.30 | 0.54 | 0.19 | 0.18 |
| $La^-$ I (mmoL·$L^{-1}$) | r | 0.45 | 0.32 | 0.42 | -0.29 | 0.42 | 0.19 |
| | p | 0.26 | 0.44 | 0.30 | 0.48 | 0.30 | 0.66 |
| $La^-$ II (mmoL·$L^{-1}$) | r | 0.24 | 0.28 | 0.29 | 0.01 | 0.54 | 0.42 |
| | p | 0.57 | 0.50 | 0.49 | 0.99 | 0.17 | 0.30 |
| $La^-$ III (mmoL·$L^{-1}$) | r | 0.14 | 0.20 | 0.19 | 0.08 | 0.01 | 0.05 |
| | p | 0.74 | 0.64 | 0.66 | 0.86 | 0.97 | 0.91 |
| $La^-$ IV (mmoL·$L^{-1}$) | r | 0.30 | 0.28 | 0.31 | 0.36 | **0.76** | **0.77** |
| | p | 0.48 | 0.51 | 0.45 | 0.38 | **0.03***  | **0.03***  |
| RPE II | r | -0.21 | -0.32 | -0.30 | 0.05 | -0.11 | -0.06 |
| | p | 0.62 | 0.43 | 0.48 | 0.90 | 0.79 | 0.89 |
| RPE III | r | -0.46 | -0.47 | -0.51 | 0.43 | 0.37 | 0.50 |
| | p | 0.25 | 0.24 | 0.20 | 0.29 | 0.37 | 0.21 |
| RPE IV | r | -0.54 | -0.34 | -0.48 | -0.37 | -0.28 | -0.40 |
| | p | 0.17 | 0.41 | 0.23 | 0.36 | 0.51 | 0.33 |

Note

* Correlation significant at $p < 0.05$. I—resting, II—after warm-up, III—before time trial, IV—after time trial. $WU_{CON}$—warm-up in water, $WU_{ARDS}$—warm-up in water with ARDS, $RE\text{-}WU_{ARDS}$—warm-up in water with application of ARDS on land during the transition phase between warm-up and swimming test.

production in the muscles compared to the control conditions. It is well known that blood lactate is not a reflective representation of muscle lactate. Solving this problem requires muscle tissue analyses, which was not performed in this study.

**Table 5. Results of the Pearson correlation (r) between time trial $t_{50}$ and between times $t_{25(1)}$, $t_{25(2)}$ and the difference Δ the parameters of the swimming cycle obtained in the ARDS protocol and RE-WU$_{ARDS}$, relative to WU$_{CON}$.**

| Variable | | Δ WU$_{ARDS}$—WU$_{CON}$ | | | Δ RE-WU$_{ARDS}$—WU$_{CON}$ | | |
|---|---|---|---|---|---|---|---|
| | | $t_{25(1)}$ | $t_{25(2)}$ | $t_{50}$ | $t_{25(1)}$ | $t_{25(2)}$ | $t_{50}$ |
| $V_{25(1)}$ (m·s$^{-1}$) | r | 0.56 | -0.13 | 0.22 | -0.15 | **0.73** | 0.50 |
| | p | 0.15 | 0.76 | 0.59 | 0.73 | **0.04*** | 0.21 |
| $V_{25(2)}$ (m·s$^{-1}$) | r | 0.20 | -0.58 | -0.22 | 0.21 | **-0.87** | -0.57 |
| | p | 0.64 | 0.14 | 0.61 | 0.62 | **0.01*** | 0.14 |
| $V_{50}$ (m·s$^{-1}$) | r | 0.25 | -0.53 | -0.17 | 0.14 | -0.22 | -0.10 |
| | p | 0.56 | 0.17 | 0.69 | 0.75 | 0.61 | 0.81 |
| $SR_{25(1)}$ (cycles·min$^{-1}$) | r | 0.35 | -0.09 | 0.14 | -0.67 | -0.53 | **-0.74** |
| | p | 0.39 | 0.84 | 0.74 | 0.07 | 0.18 | **0.04*** |
| $SR_{25(2)}$ (cycles·min$^{-1}$) | r | 0.35 | 0.11 | 0.25 | -0.59 | -0.38 | -0.58 |
| | p | 0.39 | 0.80 | 0.55 | 0.13 | 0.35 | 0.13 |
| $SL_{25(1)}$ (m·cycle$^{-1}$) | r | -0.40 | 0.01 | -0.21 | 0.50 | 0.63 | **0.73** |
| | p | 0.32 | 0.98 | 0.62 | 0.21 | 0.10 | **0.04*** |
| $SL_{25(2)}$ (m·cycle$^{-1}$) | r | -0.28 | -0.48 | -0.42 | 0.60 | -0.04 | 0.26 |
| | p | 0.51 | 0.23 | 0.30 | 0.12 | 0.93 | 0.54 |
| $SI_{25(1)}$ (m$^2$·(s·cycle)$^{-1}$) | r | -0.40 | -0.04 | -0.24 | 0.38 | 0.70 | **0.73** |
| | p | 0.32 | 0.93 | 0.58 | 0.36 | 0.05 | **0.04*** |
| $SI_{25(2)}$ (m$^2$·(s·cycle)$^{-1}$) | r | -0.11 | -0.65 | -0.43 | 0.53 | -0.36 | -0.02 |
| | p | 0.79 | 0.08 | 0.29 | 0.18 | 0.39 | 0.96 |

Note:

* Correlation significant at $p < 0.05$. I—resting, II—after warm-up, III—before time trial, IV—after time trial. WU$_{CON}$—warm-up in water, WU$_{ARDS}$—warm-up in the water with ARDS, RE-WU$_{ARDS}$—warm-up in the water with application of ARDS on land during the transition phase between warm-up and swimming test.

## Warm-up phase

Although this was the first study using a hypercapnic environment in warm-up and transition phase after warm-up, some physiological variables such as blood flow and muscle tissue analyses, that may explain responses to the strategy, were not analyzed. It was assumed that the use of ARDS during active warm-up in water would contribute to the improvement of sprint performance (50 m) in university level swimmers. The results of our study do not support this hypothesis. The explanation for the lack of changes during the swimming test of a 50 m front crawl with the use of WU$_{ARDS}$ can be the intensity of the warm-up protocol. Despite the standard protocol, the added respiratory dead space volume mask slowed the swimmers' completion time of warm-up by approximately four minutes. Using this specific mask could be negatively impacted the ability to make turns, which could change the movement pattern and speed after bouncing off the wall. In the present study, we did not examine kinematic changes during warm-up, but based on Wądrzyk et al. [50] study, we can conclude that a change in body position (which in our study was necessitated by the wearing of an ARDS mask) significantly reduces swimming speed. It is possible, that the swimmers regulated their rating of perceived exertion with a lower swimming speed during WU$_{ARDS}$ to be similar to other conditions, as indicated by the lack of statistically significant differences in RPE after this phase (1 point higher compared to others where they swam without ARDS). Too low intensity of warm-up could have been a factor limiting the achievement of adequate readiness of the body before the 50 m time trial. It is suggested, that the intensity of the warm-up before the maximal sprint effort lasting up to 30 sec should be about 70% of the difference between the anaerobic threshold and VO$_2$max intensity and last a few minutes, with a break of 30-minutes

to the main effort [51]. A properly designed warm-up and transition phase protocol should increase the concentration of lactate in the blood relative to the resting value, which in turn will not lead to greater fatigue at the end of the exercise. A lactate concentration of approximately 5 mmoL·L$^{-1}$ prior to the Wingate test improved the results of this test [51]. The presence of lactate in the muscle preparation improves muscle function by restoring the Na$^+$/K$^+$ pump whereas acidosis, accompanied by lactic acid accumulation, has no causal relationship with skeletal muscle fatigue [52]. In the current study, the lactate concentration was < 4.0 mmoL·L$^{-1}$ after warm-up, and dropped before the 50 m test to 2.24 mmoL·L$^{-1}$ (Table 2), which could be too weak a stimulus in relation to the described mechanism. Subsequent studies should consider the higher intensity of warming up in the water when swimming with ARDS.

## Re-warm-up phase

Many researchers have suggested that warm-ups should take place as close as possible to the target competition in order to have a positive effect on performance [13, 53]. As the main argument reports as sustaining an increase in body temperature after active warm-up, supporting improvements in swimming performance and reducing time up to 3%. Therefore, in the third protocol, RE-WU$_{ARDS}$, we used ARDS mask during the re-warm-up phase, which proved to be the most effective approach and improved the time of the 50 m swimming sprint. Although not statistically significant, six of the eight swimmers achieved a shorter time in the 50 m time trial during the RE-WU$_{ARDS}$ (Fig 3). In the study of Robertson et al. [15], a cycle of three apneas was used in the transition phase before the 400 m time trial that improved time in this trial, indicating that $CO_2$ breathing in the time preceding exercise may increase the body's readiness for sprint effort after an active warm-up. This is also indicated by the lower blood pH obtained during the RE-WU$_{ARDS}$ before the 50 m test (Table 4), which may be related to the changed to the right of the hemoglobin dissociation curve during acidosis according to the Bohr effect [54]. This increases the diffusion gradient of $O_2$ between capillaries and muscle cells leading to greater use of $O_2$ in cellular metabolism. As a consequence, there is a rapid dissociation of oxygen from hemoglobin in muscle cells [7].

It has been shown that the involvement of aerobic processes during maximal efforts lasting about 30 seconds is 2–26% [55]. The contribution of aerobic processes could be increased in RE-WU$_{ARDS}$, as there is a trend towards lower lactate concentration and higher concentration of bicarbonate ions after the 50 m test (in both without achieving the assumed level of statistical significance). This would especially affect the second part of the test distance. However, the results of our research do not confirm this, although the speed of the second half of the distance was the highest in these conditions. Therefore, other mechanisms to explain the improvement of swimming time under these conditions should be considered. In order to verify whether this mechanism could explain the observed differences in the current research, in-depth research is necessary e.g. during longer distances above 100 m.

## Respiratory muscle strength

It is interesting that in both warm-up protocols using ARDS, it did not lead to fatigue of the respiratory muscles after the 50 m swim trial. This suggests that breathing through the extra volume of respiratory dead space is not a limiting factor in respiratory effort, despite the altered pattern of ventilation observed at rest and moderate exercise [23]. The tube used in the current study with a volume of 1200 ml and a diameter of 2.5 cm did not seem to be a stimulus that can affect respiratory muscle fatigue. However, electromyography should be considered to confirm this in the further investigations.

## Swimming sprint performance

Varying intensity of effort during warm-up results in changes in aspects of swimming biomechanics [4]. Nepocatych et al. [56] evaluated the effect of three protocols of warm-ups on swimming performance among masters swimmers. A typical warm-up consisted of over 500 yards including at least 2x25-yard sprints at 90% of maximal effort; upper body vibrations with 100 m swim front crawl on the chest (50 yards at 40% of maximal effort and 50 yards at 90% of maximal effort) and only upper body vibrations. The results showed that there were no differences in the average stroke count when swimming a distance of 50 yards front crawl. In Balilionis et al. [54], the aim was to assess the effect of three different warm-up protocols on 50-yard swimming among collegiate swimmers: (1) no warm-up; (2) a short warm-up of 50 yards of front crawl at 40% of maximal effort and another 50 yards at 90% of maximal effort (3) the pre-start warm-up that the participant usually used. There were no significant differences in stroke count when swimming 50 yards front crawl. In this study we did not observe any differences in the kinematic parameters of the swimming cycle (distance per stroke, stroke rate, stroke index) after using different warm-up protocols. This may mean that the swimmer's activity during warm-up does not significantly affect the differences in the biomechanical variables of the swimming cycle. Based on the regression equation, we have attempted to identify relationships based on correlations between the difference in time improvement and the difference in individual parameters. We showed that only in RE-WU$_{ARDS}$ differences in the concentration of lactate ($p > 0,05$), hydrogen ions ($p > 0,05$), bicarbonate ($p > 0,05$), explain 93% of the variance. This indicates that only a comprehensive effect reveals the effect of breathing through the added respiratory dead space volume mask of 1200 ml on the improvement of swimming at 50 m time trial front crawl. Thus, this proves a direct effect on the improvement of anaerobic metabolism.

## Limitations

Several potential limitations must be considered when designing future studies of this type. The limited number of participants and their athletic level is a limiting factor of this study. However, in sports, regardless of the level of sport, the results are achieved not by the statistical significance of the test used, but by the improvement of performance [55]. It can also be considered that the outcome of this study was limited by the fact that the participants were not blinded and the results obtained were due to the occurrence of a placebo effect. A similar fact was previously reported in their work by Woorons et al. [57] investigating the effects of hypoventilation. This problem occurs with the use of ARDS, as it is impossible to conduct single or double-blind studies. However, although a psychological effect cannot be excluded in the present study, it should be noted that the subjects were not aware of or received any information about the possible impact of the tested method. However, the lack of a statistically significant improvement in time at 50 m in both protocols where ARDS was used excludes the placebo effect. It is also very interesting to investigate whether breathing through ARDS leads to a modification of the contribution of individual energy systems in securing the metabolic needs of working muscles during maximal sprint efforts.

## Future research

This study contributes to the level of knowledge available in the literature about using ARDS in swimming practice. This study can be an instruction manual for those who want to study the topic with a greater number of participants, in other cohorts (e.g. females), or in different conditions. Further study should include using the ARDS in the training protocol, regarding frequency (number of training units per week) and volume (number of intervention weeks),

which may also cause other body reactions and to provide the most appropriate individual stimulus. The changes within the design of the device increase the dead space e.g., by reducing the tube diameter, in order to induce higher respiratory resistance and monitor the respiratory gas parameters in real-time to determine changes in, among others, $PetCO_2$ should also be considered.

## Conclusions

Breathing through the added respiratory dead space volume mask of 1200 ml during the warm-up in the water or the 20 minutes passive sitting in the re-warm-up phase did not significantly affect the reduction in time of the 50 m time trial front crawl. However, the use of ARDS during the re-warm -up with a 20-minute transition phase is an effective method of improving readiness for maximal swimming effort in the 50 m time trial of front crawl, as compared to breathing through ARDS during the active warm-up in the water. Although not statistically significant, there was a trend of reduce time trial after RE-WU$_{ARDS}$. The effect obtained by using the ARDS during the re-warm-up phase can be applied in training and competitive sports conditions for swimmers to improve performance.

## Supporting information

**S1 Table. Changes in the maximal strength of the inspiratory and expiratory muscles at rest and after the end of the 50 m time trial in the tested warm-up protocols.** Note: Data presented as mean ± standard deviation. *Significant difference at $p < 0.05$. PImax $_I$—maximal inspiratory muscle strength at rest, PImax $_{IV}$—maximal inspiratory muscle strength after the test, PEmax $_I$—maximal expiratory muscle strength at rest, PEmax $_{IV}$—maximal expiratory muscle strength after the test. WU$_{CON}$—warm-up in water, WU$_{ARDS}$—warm-up in water with ARDS, RE-WU$_{ARDS}$—warm-up in the water with application of ARDS on land during the transition phase between warm-up and swimming test.
(DOCX)

## Acknowledgments

The authors would like to thank Professor Faye J. Perkins from Health and Human Performance, University of Wisconsin-River Falls, United States for her language assistance.

## Author Contributions

**Conceptualization:** Natalia Danek, Stefan Szczepan, Kamil Michalik.

**Data curation:** Natalia Danek, Kamil Michalik.

**Formal analysis:** Natalia Danek, Stefan Szczepan, Zofia Wróblewska, Kamil Michalik.

**Funding acquisition:** Natalia Danek, Stefan Szczepan, Kamil Michalik.

**Investigation:** Natalia Danek, Kamil Michalik.

**Methodology:** Natalia Danek, Kamil Michalik.

**Project administration:** Natalia Danek, Kamil Michalik.

**Resources:** Natalia Danek, Stefan Szczepan, Kamil Michalik.

**Software:** Natalia Danek, Stefan Szczepan, Zofia Wróblewska, Kamil Michalik.

**Supervision:** Natalia Danek, Marek Zatoń.

**Validation:** Natalia Danek, Stefan Szczepan, Kamil Michalik, Marek Zatoń.

**Visualization:** Natalia Danek, Stefan Szczepan.

**Writing – original draft:** Natalia Danek, Stefan Szczepan, Zofia Wróblewska, Kamil Michalik.

**Writing – review & editing:** Natalia Danek, Stefan Szczepan.

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
