## [Decision Letter · Decision Letter 0]

30 Oct 2023

PONE-D-23-30408A novel approach in determinate of the effect of using Added Respiratory Dead Space volume mask on during warm-up and re-warm-up phase on the 50 m front crawl swim performance.PLOS ONE

Dear Dr. Danek,

Thank you for submitting your manuscript to PLOS ONE. After careful consideration, we feel that it has merit but does not fully meet PLOS ONE’s publication criteria as it currently stands. Therefore, we invite you to submit a revised version of the manuscript that addresses the points raised during the review process.

Dear Authors, I would like to inform you that two reviewers have evaluated your manuscript and both have stated that it requires significant revisions. Please review the comments carefully and consider them during the revision process.

We look forward to receiving your revised manuscript.

Kind regards,

Michał Krzysztofik, Ph.D.

Academic Editor

PLOS ONE

Journal Requirements.

Additional Editor Comments:

Dear Authors,

two reviewers have evaluated your manuscript and both have stated that it requires significant revisions. Please review the comments carefully and consider them during the revision process.

Reviewers' comments:

Reviewer's Responses to Questions

**Comments to the Author**

1. Is the manuscript technically sound, and do the data support the conclusions?

Reviewer #1: Partly

Reviewer #2: Partly

2. Has the statistical analysis been performed appropriately and rigorously? 

Reviewer #1: Yes

Reviewer #2: No

3. Have the authors made all data underlying the findings in their manuscript fully available?

Reviewer #1: No

Reviewer #2: Yes

4. Is the manuscript presented in an intelligible fashion and written in standard English?

Reviewer #1: Yes

Reviewer #2: Yes

5. Review Comments to the Author

Reviewer #1: General comment:

The authors provided us an attempt to compare three warm-up routines and see if the effect of using a snorkel with added respiratory dead space would be beneficial for a 50 m maximal effort. This is an interesting approach with and a good idea to bring from the research area to the pool deck. Still, I make my reservations about what authors conclude by checking their data. To maintain the scientific coherency, I would suggest them to rethink what they want to conclude. Following you can find specific comments for improvement.

Specific comments:

The short title doesn’t match with your full title.

Line 75-77: You say that one of the main goals to warm-up under hypoxia is to increase the number of red blood cells to increase oxygen transportation. But here you used a 50 m test that is mostly done under anaerobic conditions. Do with think that this is the best justification for your research gap?

Your participants section was not written based on inclusion and exclusion criteria. Can you rewrite considering this?

Please change “body weight” by “body mass” over the text since you express the value in kg.

Line 118-120. Probably there is some formatting typo here.

I know what RBC, WBC and HGB is. But this should be described somewhere in the text.

Before using the snorkel with ARDS, you should have tested it in terms of kinematic and efficiency constraints to ensure the effort will be the same.

Probably your RE-WUARSD needed to present some intensity on dry-land near to that performed during the WUCon. Don’t you agree?

Why use 20min of passive restitution and not less or more?

Line 169-170. Some confusing was made here. If the 50 m time trial was done without the ARDS, why mentioning the device at this stage? If they not used it in the maximal test you should clarify it.

At any point of your results you show to us PO2 values. Why is that?

One of your findings is that “20 minutes of breathing through a 1200 ml added respiratory dead space volume mask during re-warm-up led to a reduction in the 50 m swim time”. I cannot agree with you, because form as statistically point of view no differences were found between the normal condition and RE-warm-up regarding the 50 m time (p = 0.9). The time differences are so tiny that can be attributed to the short effort or to the swimmers competitive level (high level swimmers are more able to maintain consistent results).

For my interpretation you cannot say that there was no “muscle fatigue”. At any point of your intervention you used any EMG measures to ensure that assumption. So, you should have some care on that.

Line 472: “unpublished”. What do you mean? This is not a scientific way to present some findings.

At the end, if the use of ARDS is more effective in a re-warm-up phase comparing to a warm-up phase. But this is not observed when comparing to the normal condition (without the use of a ARDS). At any point in your results I found difference between re-warm-up and the normal condition (except for the PCO2 at phase III, still T50 was not different). So, coaches can retain from your findings that the use of a ARDS not changes so much the swimmers physiological capacity for a 50 m effort. Do you agree? Probably in longer efforts this will have another effects on race time. Do you agree?

Reviewer #2: Dear Authors,

I would like to express my gratitude for the opportunity to review this manuscript.

The manuscript at this stage requires improvements. Below are suggestions with line indications:

1-3 – Please revise the title format, considering the journal template and instructions for authors.

45 – Please avoid abbreviations in keywords.

62-82 – This paragraph is too long. Please consider standardization in all manuscript (for example also in the discussion section) around 8-12 lines to improve readability.

113 – Please indicate FINA points.

108-113 – More information is needed about the subjects. Inclusion and exclusion criteria, routine training (water, dryland?), nutrition? Regular medicine? Informed consent? All pertinent information should be indicated

118-120 – A description of the abbreviations should be presented below the table.

130 – More than one space after the endpoint. Please correct this line and revise all manuscript.

136 – “20.01.2020 – 24.02.2020”. Please refer to possible COVID-19 effect.

167-170 – Only subjects in the swimming pool during data collection? In what lane they swam? Who collected the data (academic background and experience)? Please provide all pertinent information related to data collection.

215-221 – Different format in the subtitles. Please standardize all manuscript.

227 – Distance format is different compared for example with line 223. Please standardize in all manuscript.

249 – Please indicate sample power results considering the 8 subjects (Gpower used?).

279 – All “p” suggested in italic.

279-287 – Different decimals in the p values. Please standardize in all manuscript.

291 – Please revise the table content and format. Please make sure all abbreviations are in full in the table footnote (below the table). These details should also be considered for the following tables.

296 – Please consider including text between tables 2 and 3.

304 - Please consider including text between table 3 and figure 3.

Pages 22-28 – The discussion section is too long, please consider reducing or presenting subtopics to improve readability.

498 – Please include suggestions for future research.

538 - All references should be carefully revised, they are not according to the journal template and instructions for authors.

Please revise the format of the manuscript considering the journal template and instructions for authors. Please check all details.

Please carefully revise English details throughout the manuscript.

6. PLOS authors have the option to publish the peer review history of their article (what does this mean?). If published, this will include your full peer review and any attached files.

Reviewer #1: **Yes: **Mário J. Costa

Reviewer #2: No

---

## [Author Response · Author response to Decision Letter 0]

21 Dec 2023

Answer to Reviewer 1.

Dear Reviewer, 

thank you for your comments and suggestions. We appreciate your contribution for improving the quality of this manuscript. All changes have been made in "track the changes" mode in the main text. Below, we show detailed answers on your comments.

General comment:

The authors provided us an attempt to compare three warm-up routines and see if the effect of using a snorkel with added respiratory dead space would be beneficial for a 50 m maximal effort. This is an interesting approach with and a good idea to bring from the research area to the pool deck. Still, I make my reservations about what authors conclude by checking their data. To maintain the scientific coherency, I would suggest them to rethink what they want to conclude. Following you can find specific comments for improvement.

Specific comments:

Reviewer: The short title doesn’t match with your full title.

Answer: For a better matches, the main title has been edited to highlight hypercapnia caused by the use of the ARDS mask

Action: Change the main title: Hypercapnic warm-up and re-warm-up – a novel approach in swimming sprint Hypercapnic warm-up and re-warm-up – a novel approach in swimming sprint

Change the short title: Effects of hypercapnic warm-up in sprint swimming

Reviewer: Line 75-77: You say that one of the main goals to warm-up under hypoxia is to increase the number of red blood cells to increase oxygen transportation. But here you used a 50 m test that is mostly done under anaerobic conditions. Do with think that this is the best justification for your research gap?

Answer: We agree, it lead to misinterpretation in the introduction. 

Action: This section was shortened with following final conclusion as gap for next paragraph “It should be mentioned that Robertson et al. [15] examined swimming performance during the 400m test, in which the aerobic energy system is dominant [17]. However, for improving anaerobic swimming performance (eg. in the 50m swimming sprint) other mechanisms should be stimulated during re-warm-up phase.”

Reviewer: Your participants section was not written based on inclusion and exclusion criteria. Can you rewrite considering this?

Answer: Of course, it should be mentioned in this section. 

Action: This section was improved. We added sentence one complementary sentence ”They trained regularly eleven times per week“ and other about exclusion criteria: The exclusion criteria were as follows: 1) injuries three months before an experiment; 2) diagnosed asthma symptoms; and 3) smoking cigarettes. Additionally, we wrote some information about a priori sample size examining “The required sample size was determined by The G*Power software (version 3.1.9.2; Kiel University, Kiel, Germany).”

Reviewer: Please change “body weight” by “body mass” over the text since you express the value in kg.

Answer: Of course.

Action: Done

Reviewer: Line 118-120. Probably there is some formatting typo here.

Answer: You are right. It was changed. 

Reviewer: I know what RBC, WBC and HGB is. But this should be described somewhere in the text.

Answer: We agree, it was changed.

Action: Added “RBC - erythrocytes; WBC - leukocytes; HGB – hemoglobin concentration;” under table.

Reviewer: Before using the snorkel with ARDS, you should have tested it in terms of kinematic and efficiency constraints to ensure the effort will be the same.

Answer: During previous studies after swimming with the ARDS apparatus, subjects were individually interviewed to ask if there was any discomfort in wearing a snorkel. The authors found that during low-intensity swimming used apparatus did not affect movement's kinematic and swimming efficiency (Szczepan et al. 2020; Szczepan et al. 2020; Szczepan et al. 2022). 

Action: Improved.

Reviewer: Probably your RE-WUARSD needed to present some intensity on dry-land near to that performed during the WUCon. Don’t you agree?

Answer: No, we don’t agree with you, because the same PASSIVE break was used for all conditions. The only variable was the use of ARDS mask in the RE-WUARDS protocol, which significantly affected the change of acid-base balance, provoking the state of tolerated hypercapnia

Reviewer: Why use 20min of passive restitution and not less or more?

Answer: A 20-minute passive break based on the West et al. (2013) study was used (https://pubmed.ncbi.nlm.nih.gov/22789310/).

Reviewer: Line 169-170. Some confusing was made here. If the 50 m time trial was done without the ARDS, why mentioning the device at this stage? If they not used it in the maximal test you should clarify it.

Answer: We agree with this comment, writing in this format misleads the reader. This has been changed.

Reviewer: At any point of your results you show to us PO2 values. Why is that?

Answer: The data was supposed to be included in the supplement due to the lack of statistical differences, but for full reasoning we put the data in tables.

Action: Table 2 was enriched with pO2 analyses.

Reviewer: One of your findings is that “20 minutes of breathing through a 1200 ml added respiratory dead space volume mask during re-warm-up led to a reduction in the 50 m swim time”. I cannot agree with you, because form as statistically point of view no differences were found between the normal condition and RE-warm-up regarding the 50 m time (p = 0.9). The time differences are so tiny that can be attributed to the short effort or to the swimmers competitive level (high level swimmers are more able to maintain consistent results). At the end, if the use of ARDS is more effective in a re-warm-up phase comparing to a warm-up phase. But this is not observed when comparing to the normal condition (without the use of a ARDS). At any point in your results I found difference between re-warm-up and the normal condition (except for the PCO2 at phase III, still T50 was not different). So, coaches can retain from your findings that the use of a ARDS not changes so much the swimmers physiological capacity for a 50 m effort. Do you agree? Probably in longer efforts this will have another effects on race time. Do you agree?

Answer: We partially agree with you about overconfident inferences. However, when analyzing individual times, we observed a reduction in time in 6 out of 8 participants. Observing the reduction in swimming time by 0.3 seconds at a distance of 50 meters as a result of using one variable, which was the ARDS mask during the re-warm-up phase, it is difficult from a practical perspective as a trainer not to consider it as an effective method to improve swimming performance. Probably, the use of the RE-WUARDS phase before longer distances could cause greater effects, similar to Robertson's study, which proposed apnea phases before 400m. Future studies using ARDS are planned to be carried out at distances above 400m.

Action: We changed the part of Conclusion: Breathing through the added respiratory dead space volume mask of 1200 ml during the 20-minute passive sitting in the re-warm-up phase is an effective method of improving readiness for maximal effort in swimming in 50 m time trial front crawl, as opposed to breathing through ARDS during the active warm-up in the water. Although not statistically significant, the observed trend of reduced time trial among participants in the trials after RE-WUARDS compared to the control condition, the effect obtained during the warm-up phase can be successfully used in training conditions and sports competition of swimmers to improve performance.

Reviewer: For my interpretation you cannot say that there was no “muscle fatigue”. At any point of your intervention you used any EMG measures to ensure that assumption. So, you should have some care on that.

Answer: You are right, thank you for your suggestion.

Action: We added “Thuse, the tube used in the current study with a volume of 1200 ml and 

a diameter of 2.5 cm did not seem to be a stimulus that can affect respiratory muscle fatigue. However, electromyography should be considered to confirm this in the further investigations.”

Reviewer: Line 472: “unpublished”. What do you mean? This is not a scientific way to present some findings.

Answer: That was a mistake. It has been corrected.

Sincerely,

Authors 

Answer to Reviewer 2.

Dear Reviewer, 

thank you for your comments and suggestions. We appreciate your contribution for improving the quality of this manuscript. All changes have been made in "track the changes" mode in the main text. Below, we show detailed answers on your comments. 

Dear Authors,

I would like to express my gratitude for the opportunity to review this manuscript. The manuscript at this stage requires improvements. Below are suggestions with line indications:

Reviewer: 1-3 – Please revise the title format, considering the journal template and instructions for authors.

Answer: Thank you for your suggestions.

Action: Done.

Reviewer: 45 – Please avoid abbreviations in keywords.

Answer: Thank you for advice.

Action: It was changed.

Reviewer: 62-82 – This paragraph is too long. Please consider standardization in all manuscript (for example also in the discussion section) around 8-12 lines to improve readability.

Answer: Yes, we agree. Information about hypoventilation could be deleted. 

Action: We deleted sentences with hypoventilation and finally this paragraph is shortened. Below new version: The effect of the warm-up strategy on swimming performance also depends on the transition phase, which is the rest time between warm-up and start of the competitive event. A break longer than 20 minutes can rapidly reduce muscle temperature and impair the performance of the competition [10]. Various forms of counteracting the negative effects of passive break are used, such as wearing warm clothing [11] or using exercises on land [12], also referred to as re-warm-up procedures [13]. Using active warm-up routines and combining them with the appropriate re-warm-up strategy can effectively improve swimming performance [12]. Ramos-Campo et al. [14] showed that active exercise on land under hypoxic conditions reduced the drop in body temperature during the transition phase, thereby improving the results of the 100 m time trial of young amateur swimmers. But, provoking this condition by the specific devices in a swimming pool is not easy and may be too expensive. In contrast, Robertson et al. [15] proposed an apnea series as a warm-up component to induce hypoxia. The response to the apnea series induced also hypercapnia, decreased blood saturation, increased acidosis, bradycardia, and splenic contraction due to, among other things, hypoxemia during the initial apneic phase [16]. This leads to an increase in the number of circulating erythrocytes, suggesting a potential method to rapidly increase the body's ability to transport oxygen. It should be mentioned that Robertson et al. [15] examined swimming performance during the 400m test, in which the aerobic energy system is dominant [17]. However, for improving anaerobic swimming performance (eg. in the 50m swimming sprint) other mechanisms should be stimulated during the re-warm-up phase. 

Reviewer: 113 – Please indicate FINA points.

Answer: The sports level of the participants was determined by the best times in the 50 m freestyle competition (in the 25 m swimming pool), obtained in the last season amounting to 24.51 ± 1.37 s, categorized them as trained swimmers in their age group (565.8 ± 93.3 FINA points in a short course competition at the time of data collection).

Action: Improved.

Reviewer: 108-113 – More information is needed about the subjects. Inclusion and exclusion criteria, routine training (water, dryland?), nutrition? Regular medicine? Informed consent? All pertinent information should be indicated

Answer: Criteria for inclusion were somatic parameters of swimmers. Lines 121-124. Exclusion criteria have been added: “The exclusion criteria were as follows: 1) injuries three months before an experiment; 2) diagnosed asthma symptoms; and 3) smoking cigarettes.”

A statement is to find in Line 131: Participants were informed of the potential risks of the experiment, and all gave written informed consent prior to participation in the study. Following details have been added in Line 113(new line 162-165): Throughout the study, the individuals led a lifestyle and maintained a diet normal for people of that age. However, the participants’ diets were not controlled. All of the swimmers were reportedly free of the drugs, medication, or dietary supplements known to influence physical performance.

Line 109: (new line: 131): The most individual representative stroke were freestyle for all swimmers.

Line 135-141: An interview with the head coach revealed that during the investigation which carried on at the beginning of the winter general preparatory training period, the swimmers participated in dryland workouts and in-water practice. Swimmers trained 7-8 swimming (20-25 km) and 3 dryland sessions (2-3 hours) per week in the same squad and under the direction of the same coach. To minimize any overtraining effects on an experiment, swimmers avoided stressful training during the days before the test.

Action: Improved.

Reviewer: 118-120 – A description of the abbreviations should be presented below the table.

Answer: Of course. 

Action: It has been completed.

Reviewer: 130 – More than one space after the endpoint. Please correct this line and revise all manuscript.

Answer: Of course, sorry for this mistake.

Action: It was changed.

Reviewer: 136 – “20.01.2020 – 24.02.2020”. Please refer to possible COVID-19 effect.

Answer: In our country, patient O was reported in March 2020. We did not perform data for COVID-19 during this study.

Reviewer: 167-170 – Only subjects in the swimming pool during data collection? In what lane they swam? Who collected the data (academic background and experience)? Please provide all pertinent information related to data collection.

Answer: The measurements were taken by a specialized lab technician with a device calibrated before each trial. During data collection, the participants stayed on the first three lanes of the pool, to which others had no access.

Action: Improved in line: 214-216.

Reviewer: 215-221 – Different format in the subtitles. Please standardize all manuscript.

Answer: Of course.

Action: Done.

Reviewer: 227 – Distance format is different compared for example with line 223. Please standardize in all manuscript.

Answer: Thank you.

Action: It was changed.

Reviewer: 249 – Please indicate sample power results considering the 8 subjects (Gpower used?).

Answer: The quantitative study analysis involved a 4-dimensional approach (alpha, power, sample size, and effect size) [34] and was computed using G*Power 3.1 software (3.1.9.2, Kiel, Germany) [35]. The sample size of n=8 was set with a minimum acceptable effect size of f2=0.02 [36], with the level of significance set as a=0.05, and a power of 1-β=0.05. Effect sizes (partial eta squared (η2)) for ANOVA were interpreted as small (0.02), moderate (0.13), or large (≥0.26) [36].

Action: Improved.

Reviewer: 279 – All “p” suggested in italic.

Answer: Of course.

Action: Done.

Reviewer: 279-287 – Different decimals in the p values. Please standardize in all manuscript.

Answer: Of course.

Action: It was changed.

Reviewer: 291 – Please revise the table content and format. Please make sure all abbreviations are in full in the table footnote (below the table). These details should also be considered for the following tables.

Answer: Thank you for your suggestions.

Action: It was changed.

Reviewer: 296 – Please consider including text between tables 2 and 3.

Answer: Ok.

Action: Done.

Reviewer: 304 - Please consider including text between table 3 and figure 3.

Answer: Ok.

Action: Done.

Reviewer: Pages 22-28 – The discussion section is too long, please consider reducing or presenting subtopics to improve readability.

Answer: Thank you for this suggestion. 

Action: We have briefed some parts and organized them into sub-sections.

Reviewer: 498 – Please include suggestions for future research.

Answer: This study contributes to the level of knowledge available in the literature about using ARDSV in swimming

---

## [Decision Letter · Decision Letter 1]

13 Feb 2024

PONE-D-23-30408R1Hypercapnic warm-up and re-warm-up - a novel approach in swimming sprintPLOS ONE

Dear Dr. Danek,

Thank you for submitting your manuscript to PLOS ONE. After careful consideration, we feel that it has merit but does not fully meet PLOS ONE’s publication criteria as it currently stands. Therefore, we invite you to submit a revised version of the manuscript that addresses the points raised during the review process.

We look forward to receiving your revised manuscript.

Kind regards,

Michał Krzysztofik, Ph.D.

Academic Editor

PLOS ONE

**Additional Editor Comments:**

The article has been reviewed again, and unfortunately, the reviewers have highlighted the need for significant revisions.

Please take some time to carefully review their suggestions and comments.

Reviewers' comments:

Reviewer's Responses to Questions

**Comments to the Author**

1. If the authors have adequately addressed your comments raised in a previous round of review and you feel that this manuscript is now acceptable for publication, you may indicate that here to bypass the “Comments to the Author” section, enter your conflict of interest statement in the “Confidential to Editor” section, and submit your "Accept" recommendation.

Reviewer #1: (No Response)

Reviewer #2: All comments have been addressed

2. Is the manuscript technically sound, and do the data support the conclusions?

Reviewer #1: Partly

Reviewer #2: Partly

3. Has the statistical analysis been performed appropriately and rigorously? 

Reviewer #1: Yes

Reviewer #2: Yes

4. Have the authors made all data underlying the findings in their manuscript fully available?

Reviewer #1: Yes

Reviewer #2: Yes

5. Is the manuscript presented in an intelligible fashion and written in standard English?

Reviewer #1: Yes

Reviewer #2: No

6. Review Comments to the Author

Reviewer #1: The authors tried to make changes in the manuscript according with the reviewers’ opinion. I acknowledge their effort at some point. But for me there are some major issues that still impair the scientific quality of the work.

In the sample section you show some incoherence’s in the number of training sessions per week. Line 110 “They trained regularly 11 times per week”. Line 114 “swimmers trained 7-8 swimming and 3 dry-land sessions per week”. I understand that they reduced volume, but this can be confusing for the reader. You should choose just only one of those training backgrounds.

I don’t agree with your inclusion criteria. Sorry for my humble opinion, but I think that those are too hard that probably were not accomplished. Plus, the use of just eight swimmers is a great limitation. The use of G-Power yielding 8 subjects were based on a low power level.

I understand that “during previous studies after swimming with the ARDS apparatus, subjects

were individually interviewed to ask if there was any discomfort in wearing a snorkel”. But testing in kinematics constraints or energetic demands is bigger and more accurate than enquiry the swimmers. None of the references cited (20, 21 or 26) are abut kinematics, so you cannot write the sentence you wrote (lines 177-179). So, you are testing with a device and gathering conclusions without measuring its real impact in technique.

Even if all the conditions were under the same passive break (which I agree), this not means that all the conditions were performed under the same intensity during the effective exercise. Probably you missed to understand me.

If there were no differences in PO2 values between conditions, this means that your strategy for inducing the physiological benefits remains to be effective. Do you agree? Moreover, this goes against your description in the introduction on the Robertson et al [15] or Bakovic [16] findings.

Please keep in mind that we are not analysing this work from the trainer perspective, but form a scientific point. And if there are no differences you cannot conclude what you have written in the lines 503-509. Plus, from a scientific point I don’t know what you mean by writing “…improving readiness for maximal effort…”.

Reviewer #2: Dear Authors,

Thank you for considering my suggestions and incorporating them into the manuscript, which is globally improved, congratulations.

Below are some specific suggestions with line indication.

148 – Please revise the format of the formula.

149 – SD was previously abbreviated. Please revise these details in all manuscript.

151 – “p” suggested in italic. Please revise these details in all manuscript (another example line 344).

208, 209 – Please consider not placing the figures together in the manuscript, but immediately after their introduction in the text.

229-230 – Please revise city and country.

325 – Please revise the table 2 content and standardize the text alignment format (in this version text is centered but also at left). Please consider the same in tables 3, 4, and 5.

350-351 – Please revise.

395 – Please place space between value and unit.

417-445 – Please consider splitting the paragraph to improve readability. Same in 448-472.

416 – Without space / 447 with space / 482 different format. Please standardize.

511 – Abbreviations suggested.

586 – Please revise “ARDSV”.

599-602 – Please revise the sentence considering a clear message. It is suggested to revise all the content of the conclusions section considering the results of the study, and providing the readers with clear and direct messages, if possible, with practical application.

637 – Please revise all the references format. Some examples: Ref 1 title in uppercase, ref 2 in lowercase; Ref 22 information is missing (journal, pages, and other); Same in ref 26.

7. PLOS authors have the option to publish the peer review history of their article (what does this mean?). If published, this will include your full peer review and any attached files.

Reviewer #1: No

Reviewer #2: No

---

## [Author Response · Author response to Decision Letter 1]

15 Mar 2024

Answer to Reviewer #1: 

Reviewer: The authors tried to make changes in the manuscript according with the reviewers’ opinion. I acknowledge their effort at some point. But for me there are some major issues that still impair the scientific quality of the work.

Answer: Dear Reviewer, thank you very much for your time and contribution to improving our article. In your comment, you state that you feel that there are “some major issues that still impair the scientific quality of the work”.

Action: You gave us specific suggestions on how to address concerns, and we definitely responded to your information to strengthen the quality of our work.

Reviewer: In the sample section you show some incoherence’s in the number of training sessions per week. Line 110 “They trained regularly 11 times per week”. Line 114 “swimmers trained 7-8 swimming and 3 dry-land sessions per week”. I understand that they reduced volume, but this can be confusing for the reader. You should choose just only one of those training backgrounds.

Answer: Improved.

Action: An interview with the head coach revealed that during the investigation which was conducted at the beginning of the winter general preparatory training period, the swimmers participated in dryland workouts and in-water practice. Swimmers trained 7-8 swimming (20-25 km) and 3 dryland sessions (2-3 hours) per week in the same squad and under the direction of the same coach. During the research period, the average activity of participants was limited to 18 ± 2.5 hours of training/week.

Reviewer: I don’t agree with your inclusion criteria. Sorry for my humble opinion, but I think that those are too hard that probably were not accomplished. Plus, the use of just eight swimmers is a great limitation. The use of G-Power yielding 8 subjects were based on a low power level.

Answer: As you suggestion, the inclusion criteria needed to be more detailed. We corrected it. 

Action: The inclusion criteria were (1) at least 10 years of swimming training, (2) having the front crawl as their primary competition stroke, (3) participants were 26 years old and below (5) and able to give informed consent to participate. On the other hand, the exclusion criteria were (a) injuries three months before an experiment; (b) diagnosed asthma symptoms; and (c) smoking cigarettes. It was ‘exploratory’ research and will need further confirmation through more research with more subjects.

Reviewer: I understand that “during previous studies after swimming with the ARDS apparatus, subjects were individually interviewed to ask if there was any discomfort in wearing a snorkel”. But testing in kinematics constraints or energetic demands is bigger and more accurate than enquiry the swimmers. None of the references cited (20, 21 or 26) are abut kinematics, so you cannot write the sentence you wrote (lines 177-179). So, you are testing with a device and gathering conclusions without measuring its real impact in technique.

Answer: We agree with the reviewer on this point. 

Action: It has been removed from the text (see line 164). Is well known that devices like snorkel imposes changes in the normal biomechanical pattern when swimming front crawl. In the present case this is confirmed by Barbosa et at. (2010) Kinematical changes in swimming front Crawl and Breaststroke with the AquaTrainer® snorkel, for example. However, during a 50 m time trial the swimmers swam without ARDS apparatus (snorkel). To clarify this aspect a necessary sentence has been added to the methods section (see line 188). ARDS snorkel was used only during warm-up conditions.

Reviewer: Even if all the conditions were under the same passive break (which I agree), this does not mean that all the conditions were performed under the same intensity during the effective exercise. Probably you missed to understand me. If there were no differences in PO2 values between conditions, this means that your strategy for inducing the physiological benefits remains to be effective. Do you agree? Moreover, this goes against your description in the introduction on the Robertson et al [15] or Bakovic [16] findings.

Answer: The main factor determining the "spleen effect" is the above-mentioned hypoxia, which was the main aim the physiological response assessed in the studies by Robertson and Bakovic. The accompanying reaction was hypercapnia, which also effects of increases the number of circulating erythrocytes. However, we know that an insufficient amount of oxygen, lowering pO2, reduces the use of aerobic energy processes, giving way to the use of anaerobic sources and, consequently, increases the production of lactic acid. The compensatory mechanism we are provoking is intended to induce a state of hypercapnia, which affects the shift and delay of anaerobic glycolysis. Therefore, we agree that the lack of difference in pO2 is a desirable observation, especially when we observe a significant increase in pCO2 after the use of ARDS in the WUARDS and RE-WUARDS conditions.

Action: No action requirement

Reviewer: Please keep in mind that we are not analysing this work from the trainer perspective but form a scientific point. And if there are no differences you cannot conclude what you have written in the lines 503-509. Plus, from a scientific point I don’t know what you mean by writing “…improving readiness for maximal effort…”.

Answer: Thank you for this suggestion. We have explained this section in more detail.

Action: Breathing through the added respiratory dead space volume mask of 1200 ml during the warm-up in the water or the 20-minute passive sitting in the re-warm-up phase did not significantly affect the reduction in time of the 50 m time trial front crawl. However, the use of ARDS during the re-warm -up phase is an effective method of improving readiness for maximal swimming effort in the 50 m time trial of front crawl, as compared to breathing through ARDS during the active warm-up in the water. Although not statistically significant, there was a trend of reduce time trial after RE-WUARDS. The effect obtained by using the ARDS during the re-warm-up phase can be applied in training and competitive sports conditions for swimmers to improve performance.

Best regards, 

Authors

Answer to Reviewer #2:

Reviewer: Dear Authors, thank you for considering my suggestions and incorporating them into the manuscript, which is globally improved, congratulations. Below are some specific suggestions with line indication.

Answer: Dear Reviewer, thank you very much for your time, all comments and contributions to improving our article.

Action: Your comments have greatly improved our article. We considered every point and hope our changes and response will prove to be satisfactory.

Reviewer: 148 – Please revise the format of the formula.

Answer: Of course.

Action: Done.

Reviewer: 149 – SD was previously abbreviated. Please revise these details in all manuscript.

Answer: Of course.

Action: Done.

Reviewer: 151 – “p” suggested in italic. Please revise these details in all manuscript (another example line 344).

Answer: Of course.

Action: Done.

Reviewer: 208, 209 – Please consider not placing the figures together in the manuscript, but immediately after their introduction in the text.

Answer: According to the journal's instructions, we must upload the graphics in a separate file.

Action: No action requirement

Reviewer: 229-230 – Please revise city and country.

Answer: Of course.

Action: Done.

Reviewer: 325 – Please revise the table 2 content and standardize the text alignment format (in this version text is centered but also at left). Please consider the same in tables 3, 4, and 5.

350-351 – Please revise. 395 – Please place space between value and unit.

Answer: We apologize for these oversights.

Action: This has been corrected.

Reviewer: 417-445 – Please consider splitting the paragraph to improve readability. Same in 448-472.

Answer: Thank you.

Action: This has been changed.

Reviewer: 416 – Without space / 447 with space / 482 different format. Please standardize.

511 – Abbreviations suggested. 586 – Please revise “ARDSV”.

Answer: Of course.

Action: It was changed.

Reviewer: 599-602 – Please revise the sentence considering a clear message. It is suggested to revise all the content of the conclusions section considering the results of the study, and providing the readers with clear and direct messages, if possible, with practical application.

Answer: Thank you for this suggestion. This fragment has been edited.

Action: Breathing through the added respiratory dead space volume mask of 1200 ml during the warm-up in the water or the 20-minute passive sitting in the re-warm-up phase did not significantly affect the reduction in time of the 50 m time trial front crawl. However, the use of ARDS during the re-warm -up phase is an effective method of improving readiness for maximal swimming effort in the 50 m time trial of front crawl, as compared to breathing through ARDS during the active warm-up in the water. Although not statistically significant, there was a trend of reduce time trial after RE-WUARDS. The effect obtained by using the ARDS during the re-warm-up phase can be applied in training and competitive sports conditions for swimmers to improve performance.

Reviewer: 637 – Please revise all the references format. Some examples: Ref 1 title in uppercase, ref 2 in lowercase; Ref 22 information is missing (journal, pages, and other); Same in ref 26.

Answer: Of course.

Action: Done.

Best regards, 

Authors

---

## [Decision Letter · Decision Letter 2]

28 Aug 2024

PONE-D-23-30408R2Hypercapnic warm-up and re-warm-up - a novel approach in swimming sprintPLOS ONE

Dear Dr. Danek,

Thank you for submitting your manuscript to PLOS ONE. After careful consideration, we feel that it has merit but does not fully meet PLOS ONE’s publication criteria as it currently stands. Therefore, we invite you to submit a revised version of the manuscript that addresses the points raised during the review process.

The manuscript has improved following the review process, but some additional work is required before it can be recommended for acceptance. Please refer to the detailed comments provided by the reviewers.

We look forward to receiving your revised manuscript.

Kind regards,

Michał Krzysztofik, Ph.D.

Academic Editor

PLOS ONE

Journal Requirements:

Additional Editor Comments:

Dear Authors,

The manuscript has improved following the review process, but some additional work is required before it can be recommended for acceptance. Please refer to the detailed comments provided by the reviewers.

Reviewers' comments:

Reviewer's Responses to Questions

**Comments to the Author**

1. If the authors have adequately addressed your comments raised in a previous round of review and you feel that this manuscript is now acceptable for publication, you may indicate that here to bypass the “Comments to the Author” section, enter your conflict of interest statement in the “Confidential to Editor” section, and submit your "Accept" recommendation.

Reviewer #1: (No Response)

Reviewer #2: All comments have been addressed

Reviewer #3: All comments have been addressed

2. Is the manuscript technically sound, and do the data support the conclusions?

Reviewer #1: Partly

Reviewer #2: Partly

Reviewer #3: Partly

3. Has the statistical analysis been performed appropriately and rigorously? 

Reviewer #1: No

Reviewer #2: Yes

Reviewer #3: Yes

4. Have the authors made all data underlying the findings in their manuscript fully available?

Reviewer #1: No

Reviewer #2: Yes

Reviewer #3: Yes

5. Is the manuscript presented in an intelligible fashion and written in standard English?

Reviewer #1: Yes

Reviewer #2: Yes

Reviewer #3: Yes

6. Review Comments to the Author

Reviewer #1: In your response, you agree with me that the lower number of subjects impairs the strength of your results. First, if it is exploratory research the expression “an exploratory approach” should be in the title. Secondly, if it is exploratory research I don’t think that this should be published in a journal with a such high impact factor. But, this is my humble opinion.

Just to be accurate, Barbosa et all didn’t find any kinematical changes when swimming front crawl with the snorkel. The changes were most seen in other phases of the testing protocol like the turns and not on the stroke cycle.

Once again I have some reservations about your data and the way you make your conclusions. In this sense I leave to the editor consideration the further steps of the manuscript.

Reviewer #2: Dear Authors,

Thank you for considering my suggestions and incorporating them into the V3 of the manuscript. The document quality has been improving throughout the review process, nevertheless, some details should be revised with detail.

Below are some specific suggestions with line indication.

53 – “HR” abbreviation suggested. Please eliminate in full in line 234.

60-78 – Shorter paragraphs are suggested to improve readability. The same should be considered in 79-96 and throughout the entire manuscript.

108 – 8 “men”. “male” suggested.

133 & 149 – All “p” in italics suggested. Moreover, the space or not between symbols and values should be standardized.

154 – “regular”? Please revise all text details (format, quality of the English, and others).

261-269 – SR two times in full not necessary. Please carefully revise these details throughout the manuscript. Moreover, considering all the used equipment, please provide information regarding the manufacturer, model, city, and country.

358 & 361 – Please improve the quality of the English in these lines and throughout the manuscript.

572-575 – All references should be carefully revised and corrected. For example, in these two lines, the title format is different (upper and lowercase).

Please revise the format of the figures and tables.

Reviewer #3: Although conducted with rigor and concern for detailing all procedures, some questions are dubious or misinterpreted.

My main consideration concerns the analysis of statistical results in which the p value associated with Fischer's exact test was not significant (time in the 50 m test, for example), but post-hoc indicated a difference. Now, if the ANOVA p value (p associated with Fischer's exact test) indicates no difference, post-hoc analyzes are not even sought. It is a basic principle of statistics. Therefore, it cannot be said that there was an improvement in performance in 50 m with the hypercapnia protocol.

Furthermore, what is the purpose of testing regressions if the coefficients are not clearly presented or discussed? To perform the regressions, was the Durbin-wWatson test applied? What variable input model? Have VIF coefficients been analyzed?

7. PLOS authors have the option to publish the peer review history of their article (what does this mean?). If published, this will include your full peer review and any attached files.

Reviewer #1: No

Reviewer #2: No

Reviewer #3: **Yes: **FLAVIO ANTONIO DE SOUZA CASTRO

---

## [Author Response · Author response to Decision Letter 2]

3 Oct 2024

Answer to Reviewer #1: 

Reviewer: In your response, you agree with me that the lower number of subjects impairs the strength of your results. First, if it is exploratory research the expression “an exploratory approach” should be in the title. Secondly, if it is exploratory research I don’t think that this should be published in a journal with a such high impact factor. But, this is my humble opinion.

Answer: Dear Reviewer, thank you very much for your response. Considering the limitations of the study due to the low number of participants and possible concerns about inference, we would like to inform you that more detailed statistical analyses were performed. At this stage of the study, we do not have the ability to increase the number of participants hence, as suggested, we have included in the title information about exploratory research “an exploratory approach” 

Action: change title to “Hypercapnic warm-up and re-warm-up: a novel exploratory approach in swimming sprint”

Reviewer: Just to be accurate, Barbosa et all didn’t find any kinematical changes when swimming front crawl with the snorkel. The changes were most seen in other phases of the testing protocol like the turns and not on the stroke cycle. Once again I have some reservations about your data and the way you make your conclusions. In this sense I leave to the editor consideration the further steps of the manuscript.

Answer: thank you for this comment. We explain our apologies for the inaccurate explanation. In the article it was emphasized that the ARDS mask was used only in warm-up techniques which could affect kinematic changes during swimming. We did not study this during the warm-up, hence the citation of the study by Barbobsy et al 2010, which did not show changes in movement kinematics during the 100m swim In our study we did not use ARDS in the 50 m time trial front crawl swimming performance. This fragment was prominently displayed to avoid misleading the reader in line 164-165.

Action: Added in line: 164-165: The ARDS mask was used only in warm-up techniques in water or on land; it was not used in the 50 m time trial front crawl swimming performance.

Best regards, 

Authors

Answer to Reviewer #2: 

Reviewer: Dear Authors,

Thank you for considering my suggestions and incorporating them into the V3 of the manuscript. The document quality has been improving throughout the review process, nevertheless, some details should be revised with detail.

Answer: Thank you for recognizing the work put into improving this manuscript. 

Below are some specific suggestions with line indication.

Reviewer: 53 – “HR” abbreviation suggested. Please eliminate in full in line 234.

60-78 – Shorter paragraphs are suggested to improve readability. The same should be considered in 79-96 and throughout the entire manuscript.

108 – 8 “men”. “male” suggested.

133 & 149 – All “p” in italics suggested. Moreover, the space or not between symbols and values should be standardized.

Answer: Thank you for suggestions.

Action: This has been changed.

Reviewer: 154 – “regular”? Please revise all text details (format, quality of the English, and others).

Answer: The article has been proofread by a native speaker.

Action: The article has been linguistically improved.

Reviewer: 261-269 – SR two times in full not necessary. Please carefully revise these details throughout the manuscript. Moreover, considering all the used equipment, please provide information regarding the manufacturer, model, city, and country.

Answer: Thank you for the suggestions.

Action: Done.

Reviewer: 358 & 361 – Please improve the quality of the English in these lines and throughout the manuscript.

Answer: The article has been proofread by a native speaker.

Action: The article has been linguistically improved.

Reviewer: 572-575 – All references should be carefully revised and corrected. For example, in these two lines, the title format is different (upper and lowercase).

Answer: Of course, thank you for this suggestion. 

Action: Done. 

Reviewer: Please revise the format of the figures and tables.

Answer: Thank you for the suggestions.

Action: This has been verified.

Best regards, 

Authors

Answer to Reviewer #3: 

Reviewer: Although conducted with rigor and concern for detailing all procedures, some questions are dubious or misinterpreted. My main consideration concerns the analysis of statistical results in which the p value associated with Fischer's exact test was not significant (time in the 50 m test, for example), but post-hoc indicated a difference. Now, if the ANOVA p value (p associated with Fischer's exact test) indicates no difference, post-hoc analyzes are not even sought. It is a basic principle of statistics. Therefore, it cannot be said that there was an improvement in performance in 50 m with the hypercapnia protocol.

Answer: Thank you for your insightful comments on our manuscript regarding the presentation of statistical measures in Table 2 and 3. We appreciate your attention to detail and your suggestions for improving the clarity and robustness of our data presentation. Upon reviewing your comments, we would like to provide some rationale for our choice to maintain the presentation of results. We removed the analysis of interactions between conditions from Table 2 and Table 3. If the analyses showed statistical significance in the main effect, we performed a detailed analysis of the interaction between the conditions and the results were described in the text (line 354-399). We believe that these measures hold a stronger relevance to the practical aspects of swimming and are more comprehensible to the target audience of this scientific communication. In conclusion, we appreciate your suggestion, on the basis of which we have corrected the presented results, so that, despite the lack of statistical significance of some parameters, we present a trend of changes useful to the practical needs and level of understanding of our target group in the swimming community. Every effort has been made to clarify in the text that the effects of using the ARDS mask in the various stages of the warm-up should be interpreted with caution. Thank you again for your insightful review and we remain open to any further suggestions and feedback.

Action: Extensive changes have been made to the results and disscussion sections. 

Reviewer: Furthermore, what is the purpose of testing regressions if the coefficients are not clearly presented or discussed? 

Answer: This study has also demonstrated several factors can potentially explain the time of performance The strongest predictor variables of time trial time t25(2) in the regression model were La-, HCO3-, H+ (p<0.05), and the strength of the relationship between model and the dependent variable was at the 93% level.

Action: The regressions analysis descriptions has been added to the statistical analysis, results, and discussion section.

Reviewer: To perform the regressions, was the Durbin-Watson test applied? 

Answer: The Durbin-Watson value has been added into the text.

Action: See sections: Statistical analyses/Results.

Reviewer: What variable input model? Have VIF coefficients been analyzed?

Answer: The information regarding to dependent (explained) variables and independent variables (explanatory) was applied into the text. The VIF value has been added as well.

Action: See sections: Statistical analyses/Results.

Best regards, 

Authors

---

## [Editor Report · Decision Letter 3]

6 Nov 2024

Hypercapnic warm-up and re-warm-up – a novel experimental approach in swimming sprint

PONE-D-23-30408R3

Dear Dr. Danek,

We’re pleased to inform you that your manuscript has been judged scientifically suitable for publication and will be formally accepted for publication once it meets all outstanding technical requirements.

Kind regards,

Michał Krzysztofik, Ph.D.

Academic Editor

PLOS ONE

Additional Editor Comments (optional):

The authors have made the necessary adjustments.
---

## [Editor Report · Acceptance letter]

14 Nov 2024

PONE-D-23-30408R3 

PLOS ONE

Dear Dr. Danek, 

I'm pleased to inform you that your manuscript has been deemed suitable for publication in PLOS ONE. Congratulations! Your manuscript is now being handed over to our production team.

Kind regards, 

on behalf of

Dr. Michał Krzysztofik 

Academic Editor

PLOS ONE